# BIASES IN EVALUATION OF MOLECULAR OPTIMIZATION METHODS AND BIAS REDUCTION STRATEGIES

## ABSTRACT

We are interested in *in silico* evaluation methodology for molecular optimization methods. Given a sample of molecules and their properties of our interest, we wish not only to train a generator of molecules that can find those optimized with respect to a target property but also to evaluate its performance accurately. A common practice is to train a predictor of the target property on the sample and use it for both training and evaluating the generator. We theoretically investigate this evaluation methodology and show that it potentially suffers from two biases; one is due to misspecification of the predictor and the other to reusing the same sample for training and evaluation. We discuss bias reduction methods for each of the biases, and empirically investigate their effectiveness.

## 1 INTRODUCTION

Molecular optimization aims to discover novel molecules with improved properties, which is often formulated as reinforcement learning by modeling the construction of a molecule using a Markov decision process. The performance of such agents is measured by the quality of generated molecules. In the community of machine learning, most of the molecular optimization methods have been verified by computer simulation. Since most of the generated molecules are novel, their properties are unknown and we have to resort to a predictor to estimate the properties. However, little attention has been paid to how reliable such estimates are, except for a few empirical studies (Renz et al., 2019; Langevin et al., 2022), making the existing performance estimates less reliable. In this paper, we study the statistical properties of such performance estimators to enhance our understanding of the evaluation protocol and we discuss several directions to improve it.

Let us first introduce a common practice to estimate the performance. Let $\mathcal{S}^\star$ be a set of molecules, $f^\star \colon \mathcal{S}^\star \to \mathbb{R}$ be a *property function* evaluating the target property of the input molecule, and $\mathcal{D} = \left\{(m_n, f^\star(m_n)) \in \mathcal{S}^\star \times \mathbb{R}\right\}_{n=1}^N$ be a sample. We typically train a predictor $f(m; \mathcal{D})$ using $\mathcal{D}$, regard it as the true property function, and follow the standard evaluation protocol of online reinforcement learning. That is, an agent is trained so as to optimize the properties of discovered molecules computed by $f(m; \mathcal{D})$, and its performance is estimated by letting it generate novel molecules and estimating their properties by $f(m; \mathcal{D})$. We call this a *plug-in performance estimator* (section 2.1).

Our research question is *how accurate the plug-in performance estimator is as compared to the true performance computed by $f^\star$*. We first point out that the plug-in performance estimator is biased in two ways, indicating that it is not reliable in general (section 2.2). The first bias called a *model misspecification bias* comes from the deviation between the predictor and the true property function evaluated over the molecules discovered by the learned agent. This bias is closely related to the one encountered in covariate shift (Shimodaira, 2000). It grows if molecules discovered by the agent become dissimilar to those used to train the predictor. The second bias called a *reusing bias* is caused by reusing the same dataset for training and testing the agent. Due to these biases, the plug-in performance estimator is not necessarily a good estimator of the true performance.

We then discuss strategies to reduce these two biases. Section 3.1 introduces three approaches to reducing the misspecification bias. Since it is caused by covariate shift, it can be reduced by training the predictor taking it into account (section 3.1.1) and/or by constraining the agent so that the generated molecules become similar to those in the sample (section 3.1.2). Yet another approach is to use a more sophisticated estimator called a doubly-robust performance estimator (section 3.1.3).

Our idea to correct the reusing bias comes from the analogy to model selection (Konishi & Kitagawa, 2007), whose objective is to estimate the test performance by correcting the bias of the training performance, *i.e.*, the performance computed by reusing the same dataset for training and testing. Given the analogy, one may consider train-test split could be the first choice. We however argue that it is not as effective as that applied to model selection due to the key difference between our setting and model selection; the test set in model selection is used to take expectation, while that in our setting is used to train a predictor, which is much more complex than expectation. This complexity introduces a non-negligible bias to the train-test split estimator, resulting in a less accurate bias estimation (section 3.2.1). We instead propose to use a bootstrap method in section 3.2.2, which is proven to estimate the reusing bias more accurately than the train-test split method does.

We empirically validate our theory in section 4. First, we quantify the two biases, and confirm that both of them are non-negligible, and the reusing bias increases as the sample size decreases, as predicted by our theory. Second, we assess the effectiveness of the bias reduction methods, and confirm that the reusing bias can be corrected, while the misspecification bias can be reduced but at the cost of performance degradation of the agent.

**Notation.** For any distribution $G$, let $\hat{G} \sim G^N$ denote the empirical distribution of a sample of $N$ items independently drawn from $G$. For a set $\mathcal{X}$, let $\delta_x$ be Dirac's delta distribution at $x \in \mathcal{X}$. For any integer $M \in \mathbb{N}$, let $[M] := \{0, \dots, M-1\}$. For any set $A$, $\mathcal{P}(A)$ denotes the set of probability distributions defined over $A$.

**Problem setting.** We define a molecular optimization problem using a Markov decision process (MDP) of length $H + 1$ ($H \in \mathbb{N}$). See appendix A for concrete examples. Let $\mathcal{S}$ be a set of states, and $s_\perp \in \mathcal{S}$ be the terminal state. Let $\mathcal{S}^\star \subseteq \mathcal{S}$ be a subset of states that correspond to valid molecules and the rest of the states correspond to possibly incomplete representations of molecules (*invalid molecules*). Let $\mathcal{A}$ be a set of actions that transform a valid or invalid molecule into another one. There exists the terminal action $a_\perp \in \mathcal{A}$ that evaluates the property of the molecule at step $H$, after which the state transits to the terminal state $s_\perp$. For each step $h \in [H+1]$, let $T_h: \mathcal{S} \times \mathcal{A} \to \mathcal{P}(\mathcal{S})$ be a state transition distribution, $r_h: \mathcal{S} \times \mathcal{A} \to \mathbb{R}$ be a reward function, and $\rho_0 \in \mathcal{P}(\mathcal{S})$ be the initial state distribution. We assume that the set of states at step $H$ is limited to $\mathcal{S}^\star$, and the reward function is defined as $r_h(s, a) = 0$ for $h \in [H]$ and $r_H(s, a_\perp) = f^\star(s)$ for $s \in \mathcal{S}^\star$. Let $\mathcal{M} = \{\mathcal{S}, \mathcal{A}, \{T_h\}_{h=0}^H, \rho_0, H\}$ be the dynamical model of the MDP. Throughout the paper, we assume we know $\mathcal{M}$ and omit the dependency on it in expressions.

Let $\Pi$ be the set of policies and $\pi = \{\pi_h(\cdot \mid s)\}_{h=0}^H \in \Pi$ be a policy modeled by a probability distribution over $\mathcal{A}$ conditioned on $s \in \mathcal{S}$. At each step $h \in [H+1]$, the agent takes action $a_h$ sampled from $\pi_h(\cdot \mid s_h)$. The performance of a policy is measured by the expected cumulative reward, $J^\star(\pi) := \mathbb{E}^\pi[\sum_{h=0}^H r_h(S_h, A_h)] = \mathbb{E}^\pi[f^\star(S_H)]$, where $\mathbb{E}^\pi[\cdot]$ is the expectation with respect to the Markov process induced by applying policy $\pi$ on $\mathcal{M}$. Letting $p_h^\pi \in \mathcal{P}(\mathcal{S}^\star)$ be the distribution of states visited by policy $\pi$ at step $h \in [H+1]$, the expected cumulative reward is alternatively expressed as, $J^\star(\pi) = \mathbb{E}_{S \sim p_H^\pi} f^\star(S)$.

In practice, the property function is not available and instead a sample from it, $\mathcal{D} = \{(m_n, f^\star(m_n)) \in \mathcal{S}^\star \times \mathbb{R}\}_{n=1}^N$, is available. Let us assume that each tuple is independently distributed according to $G \in \mathcal{P}(\mathcal{S}^\star \times \mathbb{R})$. Let $G_S \in \mathcal{P}(\mathcal{S}^\star)$ be the marginalized distribution over $\mathcal{S}^\star$ induced from $G$. For a theoretical reason clarified in appendix B, we use the empirical distribution of the sample, $\hat{G} \in \mathcal{P}(\mathcal{S}^\star \times \mathbb{R})$, rather than the sample itself (assumption 10) and we call $\hat{G}$ an empirical distribution and a sample interchangeably.

Let us define a *policy learner* $\alpha_\pi: \mathcal{P}(\mathcal{S}^\star \times \mathbb{R}) \to \Pi$, an algorithm to learn a policy from a distribution over $\mathcal{S}^\star \times \mathbb{R}$. It typically receives a sample $\hat{G}$ and outputs a policy, which we denote $\hat{\pi} := \alpha_\pi(\hat{G})$. Our objective is to evaluate its performance $J^\star(\hat{\pi})$ only given access to $\alpha_\pi$, $\hat{G}$, and $\mathcal{M}$.

## 2 BIASES OF PLUG-IN PERFORMANCE ESTIMATOR

A widely used approach to estimating $J^\star(\hat{\pi})$ is a *plug-in performance estimator* (section 2.1). We point out that it is biased in two ways (section 2.2) and theoretically characterize these biases in sections 2.3 and 2.4.

## 2.1 PLUG-IN PERFORMANCE ESTIMATOR

For any function $f : \mathcal{S}^\star \to \mathbb{R}$ and policy $\pi$, let us define a *plug-in performance function*,

$$J_{\mathrm{PI}}(\pi, f) := \mathbb{E}^\pi[f(S_H)]. \tag{1}$$

Let $\alpha_f : \mathcal{P}(\mathcal{S}^\star \times \mathbb{R}) \to (\mathcal{S}^\star \to \mathbb{R})$ be an algorithm to learn a predictor, typically by minimizing the loss function averaged over the input distribution. Let $\hat{\pi} = \alpha_\pi(\hat{G})$ be a policy trained using $\hat{G}$ and $\hat{f} := \alpha_f(\hat{G})$ be a predictor trained using the same $\hat{G}$. Then, the *plug-in performance estimator* is defined as $J_{\mathrm{PI}}(\hat{\pi}, \hat{f})$, which is often used as a proxy for the true performance, $J^\star(\hat{\pi})$.

## 2.2 BIAS DECOMPOSITION

The plug-in performance estimator is biased in two ways; the first bias comes from model misspecification of the predictor, and the second one is due to reusing the same sample for learning a policy and a predictor. Let us define $\tilde{J}_{\mathrm{PI}}(G_1, G_2) := J_{\mathrm{PI}}(\alpha_\pi(G_1), \alpha_f(G_2))$ and $\Delta_{\mathrm{PI}}(G_1, G_2) := \tilde{J}_{\mathrm{PI}}(G_1, G_2) - J^\star(\alpha_\pi(G_1))$. The quantity $\tilde{J}_{\mathrm{PI}}(G_1, G_2)$ denotes the estimated performance of a policy trained with distribution $G_1$ evaluated by a predictor trained with $G_2$, and $\Delta_{\mathrm{PI}}(G_1, G_2)$ denotes the deviation of the estimated performance from the ground truth. Then, the bias we care is denoted by $\mathbb{E}_{\hat{G} \sim G^N} \Delta_{\mathrm{PI}}(\hat{G}, \hat{G})$, which is decomposed as shown in theorem 1.

**Theorem 1.** *The bias is decomposed into a* reusing bias *and a* misspecification bias *as follows:*

$$\begin{aligned}
\mathbb{E}_{\hat{G} \sim G^N} \Delta_{\mathrm{PI}}(\hat{G}, \hat{G}) &= \mathbb{E}_{\hat{G} \sim G^N}[\tilde{J}_{\mathrm{PI}}(\hat{G}, \hat{G}) - \tilde{J}_{\mathrm{PI}}(\hat{G}, G) + \tilde{J}_{\mathrm{PI}}(\hat{G}, G) - J^\star(\hat{\pi})] \\
&= \underbrace{\mathbb{E}_{\hat{G} \sim G^N}[\tilde{J}_{\mathrm{PI}}(\hat{G}, \hat{G}) - \tilde{J}_{\mathrm{PI}}(\hat{G}, G)]}_{\text{Reusing bias}} + \underbrace{\mathbb{E}_{\hat{G} \sim G^N} \Delta_{\mathrm{PI}}(\hat{G}, G)}_{\text{Misspecification bias}}. 
\end{aligned} \tag{2}$$

## 2.3 MISSPECIFICATION BIAS

Letting $f^\infty := \alpha_f(G)$, the squared misspecification $\Delta_{\mathrm{PI}}(\hat{G}, G)^2$ is upperbounded by Jensen's inequality as,

$$\Delta_{\mathrm{PI}}(\hat{G}, G)^2 = \left(\mathbb{E}^{\hat{\pi}}(f^\infty(S_H) - f^\star(S_H))\right)^2 \le \mathbb{E}_{S \sim p_H^{\hat{\pi}}}(f^\infty(S) - f^\star(S))^2. \tag{3}$$

Assuming that $f^\infty = \operatorname{argmin}_f \mathbb{E}_{S \sim G_S}(f(S) - f^\star(S))^2$ holds, the bias increases if $f^\infty$ fails to predict the properties of molecules generated by policy $\hat{\pi}$, which occurs when the predictor is misspecified (*i.e.*, $f^\infty \ne f^\star$) and $p_H^{\hat{\pi}}$ and $G_S$ are largely deviated (*i.e.*, the discovered molecules are not similar to those in the sample).

## 2.4 REUSING BIAS

The former term of equation 2,

$$b_{\mathrm{PI}}^N(G) := \mathbb{E}_{\hat{G} \sim G^N}[\tilde{J}_{\mathrm{PI}}(\hat{G}, \hat{G}) - \tilde{J}_{\mathrm{PI}}(\hat{G}, G)], \tag{4}$$

quantifies the bias caused by reusing the same finite sample for training and testing a policy, which we call a reusing bias[1].

Let us theoretically analyze the reusing bias, assuming the sample size $N$ is moderately large such that the asymptotic expansions are valid but $O(1/N)$ term cannot be ignored. We show in proposition 2 that the reusing bias is $O(1/N)$. See appendix B for the assumptions and appendix D.2 for its proof.

**Proposition 2.** *Under assumptions 10 and 12,*

$$b_{\mathrm{PI}}^N(G) = \frac{1}{2N} \mathbb{E}_{X \sim G} \left[ 2\tilde{J}_{G,G}^{(1,1)}(\delta_X - G, \delta_X - G) + \tilde{J}_{G,G}^{(0,2)}(\delta_X - G, \delta_X - G) \right] + O(1/N^2),$$

*holds, indicating that $b_{\mathrm{PI}}^N(G) = O(1/N)$ where $\tilde{J}_{G,G}^{(1,1)}$ and $\tilde{J}_{G,G}^{(0,2)}$ are the $(1,1)$-st and $(0,2)$-nd Fréchet derivative of $\tilde{J}_{\mathrm{PI}}(G_1, G_2)$ at $(G_1, G_2) = (G, G)$.*

---

[1]The reusing bias is caused by sample reuse as well as the finiteness of the sample, which is clear when the policy is independent from $\hat{G}$; the reusing bias still exists in such a case if $\hat{f} \ne f^\infty$.

In particular, if the policy is optimal and the estimated property function is unbiased, *i.e.*, $\mathbb{E}_{\hat{G}\sim G^N}\hat{f} = f^\infty$ (which is true at least for a linear model), we can prove that the bias is optimistic (proposition 3). See appendix E for its proof.

**Proposition 3.** *Assume* $\mathbb{E}_{\hat{G}\sim G^N}\hat{f} = f^\infty$ *and* $\hat{\pi} = \operatorname{argmax}_{\pi\in\Pi} J_{\text{PI}}(\pi, \hat{f})$ *hold. Then,* $b^N_{\text{PI}}(G) \geq 0$.

## 3 BIAS REDUCTION STRATEGIES

We have witnessed that the plug-in performance estimator is biased in two ways. In this section, we discuss how to reduce these biases to obtain reliable performance estimates.

### 3.1 REDUCING MISSPECIFICATION BIAS

There are mainly three approaches to reducing the misspecification bias, $\Delta_{\text{PI}}(\hat{G}, G)$. The first one is to train the predictor considering the *covariate shift*, a mismatch between training and testing distributions (section 3.1.1). The second approach is to constrain a policy such that the molecules discovered by the policy become similar to those in the sample $\hat{G}$ (section 3.1.2). These are mainly motivated by minimizing the right-hand side of equation 3. The third one is motivated by a standard technique in contextual bandit, the *doubly-robust performance estimator* instead of the plug-in performance estimator (section 3.1.3).

Before going into details, let us introduce the notion of importance weight, which is used extensively to reduce the misspecification bias. Let $F \in \mathcal{P}(\mathcal{S}^\star)$ be any probability distribution over molecules whose support is larger than that of $p^\pi_H$. Let $(p^\pi_H/F)(s) := p^\pi_H(s)/F(s)$ ($s \in \mathcal{S}^\star$) denote the importance weight between them, and let $\alpha_w \colon \Pi \times \mathcal{P}(\mathcal{S}^\star) \to (\mathcal{S}^\star \to \mathbb{R}_{\geq 0})$ denote an algorithm that receives a policy and a distribution over molecules and outputs the importance weight between the state distribution induced by the policy and the distribution. We typically use the algorithm by substituting sample $\hat{G}$ from $G$ for the distribution, expecting that $\alpha_w(\pi, \hat{G}) \approx p^\pi_H/G$.

#### 3.1.1 COVARIATE SHIFT

The misspecification bias can be reduced by minimizing the right-hand side of equation 3, which is the mean squared error over $S \sim p^{\hat{\pi}}_H$. The predictor $f^\infty$ is usually trained by minimizing $\mathbb{E}_{S\sim G_S}(f(S) - f^\star(S))^2$ with respect to $f$ and does not necessarily minimize the right-hand side of equation 3 due to covariate shift (Shimodaira, 2000), *i.e.*, the mismatch between the training and testing distributions. One approach suggested by the author to alleviating it is to train the predictor by weighted maximum-likelihood estimation. Let us define the algorithm as,

$$\alpha^\lambda_f(w, G) = \operatorname*{argmin}_{f\in\mathcal{F}} \mathbb{E}_{S\sim G} w(S)^\lambda (f(S) - f^\star(S))^2, \tag{5}$$

where $w$ is any importance weight and $\lambda \in [0, 1]$ controls the bias and variance of the estimated predictor[2]. By substituting $\alpha^\lambda_f(w, G)$ for $\alpha_f(G)$, the misspecification bias will be reduced.

#### 3.1.2 CONSTRAIN A POLICY

The first approach does not always work. If $p^{\hat{\pi}}_H$ and $G$ are not close enough, the effective sample size of the weighted maximum-likelihood estimation becomes small, leading to poor estimation. This suggests that not all policy learners can be accurately evaluated; those whose state distribution $p^\pi_H$ is deviated from $G$ are difficult to be evaluated.

Let us assume that the policy is obtained by solving the following optimization problem: $\alpha_\pi(G) = \operatorname{argmin}_{\pi\in\Pi} \ell(\pi; G)$. While a natural approach is to add a divergence between the generator and the data distribution $P$ to the objective function as a regularization term, it is computationally expensive, especially when the length of MDP, $H$, is large. We instead propose to regularize the policy, inspired by *behavior cloning* (Fujimoto & Gu, 2021). Let us first introduce behavior cloning, and then, discuss how to apply its idea to our problem setting.

---

[2]While $\lambda = 1$ is optimal for $N \to \infty$, it will increase the variance for a finite sample size $N$, and a smaller $\lambda$ is favored.

Behavior cloning regularizes the policy so that the policy imitates a *behavior policy* that generates the data. Let us assume that there exists a behavior policy $\pi_{\mathrm{b}}$ that induces the data distribution, *i.e.*, $p_H^{\pi_{\mathrm{b}}}(s) = G(s)$ for $s \in \mathcal{S}^\star$, which may not be available in our setting. Behavior cloning employs the following regularized objective function: $\ell(\pi; G) - \frac{\nu}{H+1} \sum_{h=0}^{H} \mathbb{E}_{S_h \sim p_h^{\pi_{\mathrm{b}}}, A_h \sim \pi_{\mathrm{b}}(S_h)} [\log \pi(A_h \mid S_h)]$, where $\nu \geq 0$ is a hyperparameter controlling behavior cloning. The larger $\nu$ is, the more the learned policy resembles the behavior policy, which in turn will make $p_H^\pi$ close to the data distribution, and thus, we expect to reduce the misspecification bias.

A technical challenge in applying behavior cloning to our setting is that $\pi_{\mathrm{b}}$ is not available. Our key observation to this challenge is that while $\pi_{\mathrm{b}}$ is not available, it is often the case that a trajectory towards each molecule in the dataset can be reconstructed. For example, in an MDP that constructs a molecule atom-wisely (You et al., 2018), such a trajectory is easily obtained by removing atoms one by one from the molecule; in another MDP that constructs a molecule by chemical reactions (Gottipati et al., 2020), since each molecule in the dataset is assumed to be synthesizable (because the molecules in the dataset do exist in reality and thus are synthesizable), such a trajectory is easily obtained at least for those molecules in the dataset. Letting $\pi_{\mathrm{b}}^{-1}(m) = (s_0, a_0, s_1, a_1, \ldots, s_H = m)$ be a (potentially random) function to reconstruct a trajectory from a molecule, we propose to train a policy with regularization to the data distribution by the following optimization problem:

$$\alpha_\pi^\nu(G) := \operatorname*{argmin}_{\pi \in \Pi} \ell(\pi; G) - \frac{\nu}{H+1} \sum_{h=0}^{H} \mathbb{E}_{M \sim G} \mathbb{E}_{S_0, A_0, \ldots, S_H \sim \pi_{\mathrm{b}}^{-1}(M)} [\log \pi(A_h \mid S_h)]. \quad (6)$$

Given the discussion above, at least $\alpha_\pi^\nu(\hat{G})$ can be computed. Although this regularization is not sufficient to constrain the divergence between $p_H^{\hat{\pi}}$ and $G$ (which has been discussed in the literature of imitation learning), we consider the idea of behavior cloning is a simple yet effective heuristic, which will be investigated in the experiment.

### 3.1.3 DOUBLY-ROBUST PERFORMANCE ESTIMATOR

The third approach to reducing the misspecification bias is a *doubly-robust performance estimator*, which has been applied in contextual bandit (Dudík et al., 2014) and offline reinforcement learning (Tang et al., 2020) as an alternative to the plug-in performance estimator. Noticing that the performance can also be estimated via importance sampling, which we call an *importance-sampling performance estimator*, the doubly-robust performance estimator combines these two estimators so as to inherit their benefits.

**Importance-Sampling Performance Estimator.** Given that $J^\star(\pi) = \mathbb{E}^\pi f^\star(S_H) = \mathbb{E}_{S \sim G_S}(p_H^\pi/G_S)(S)f^\star(S)$ holds, we obtain the importance-sampling performance estimator by substituting an importance weight model for the true importance weight. For any importance weight $w \colon \mathcal{S}^\star \to \mathbb{R}_{\geq 0}$ and distribution $F \in \mathcal{P}(\mathcal{S}^\star \times \mathbb{R})$, let us define an *importance-sampling performance function* as, $J_{\mathrm{IS}}(w, F) := \mathbb{E}_{S \sim F_S} w(S) f^\star(S)$. Then, we obtain the *importance-sampling performance estimator* as $J_{\mathrm{IS}}(\hat{w}, \hat{G})$, where $\hat{w} := \alpha_w(\hat{\pi}, \hat{G})$.

**Doubly-Robust Performance Estimator.** The *doubly-robust performance function* combines the plug-in and importance-sampling performance functions as follows:

$$J_{\mathrm{DR}}(\pi, w, f, F) := \mathbb{E}_{S \sim F_S} [w(S)(f^\star(S) - f(S))] + \mathbb{E}^\pi f(S_H). \quad (7)$$

This performance function is a combination of the two performance functions in that it is related to them as, $J_{\mathrm{DR}}(\pi, 0, f, F) = J_{\mathrm{PI}}(\pi, f)$ and $J_{\mathrm{DR}}(\pi, w, 0, F) = J_{\mathrm{IS}}(w, F)$. By substituting $\hat{\pi}, \hat{w}, \hat{f}$, and $\hat{G}$ for the arguments, we obtain the doubly-robust performance estimator as $J_{\mathrm{DR}}(\hat{\pi}, \hat{w}, \hat{f}, \hat{G})$. Let us define, $\tilde{J}_{\mathrm{DR}}(G_1, G_2) := J_{\mathrm{DR}}(\alpha_\pi(G_1), \alpha_w(\alpha_\pi(G_1), G_2), \alpha_f(G_2), G_2)$. Then, the misspecification bias is expressed as, $\Delta_{\mathrm{DR}}(\hat{G}, G) := \tilde{J}_{\mathrm{DR}}(\hat{G}, G) - J^\star(\hat{\pi}) = \mathbb{E}_{S \sim G_S}(w^\infty(S) - (p_H^{\hat{\pi}}/G)(S))(f^\star(S) - f^\infty(S))$, where $w^\infty := \alpha_w(\hat{\pi}, G)$. This suggests that the misspecification bias disappears if the predictor or the importance weight is well-specified.

**Discussion.** Notice that the misspecification biases of $J_{\mathrm{PI}}$ and $J_{\mathrm{IS}}$ are given by the followings:

$$\Delta_{\mathrm{PI}}(\hat{G}, G) = J_{\mathrm{PI}}(\hat{\pi}, f) - J^\star(\hat{\pi}) = \mathbb{E}_{S \sim G_S} \left[ (p_H^{\hat{\pi}}/G)(s)(f^\infty(S) - f^\star(S)) \right],$$

$$\Delta_{\mathrm{IS}}(\hat{G}, G) := J_{\mathrm{IS}}(\hat{w}, G) - J^\star(\hat{\pi}) = \mathbb{E}_{S \sim G_S} \left[ (w^\infty(S) - (p_H^{\hat{\pi}}/G)(S))f^\star(S) \right].$$

We can deduce that for $S \sim G_S$ (i) if $|f^\star(S) - f^\infty(S)| \ll |f^\star(S)|$ holds, the misspecification bias of $J_{\mathrm{DR}}$ will be smaller than that of $J_{\mathrm{IS}}$, and (ii) if $|w^\infty(S) - (p_H^{\tilde{\pi}}/G)(S)| \ll |(p_H^{\tilde{\pi}}/G)(S)|$ holds, the misspecification bias of $J_{\mathrm{DR}}$ will be smaller than that of $J_{\mathrm{PI}}$. Therefore, if we can learn both of the predictor and the importance weight well, the doubly-robust performance estimator is preferred to the others. Otherwise, the doubly-robust one can be worse than the others.

### 3.1.4 SUMMARY

We have introduced three approaches to reducing misspecification bias. The first one trains the predictor by weighted maximum likelihood estimation (equation 5). The second one constrains the policy by behavior cloning (equation 6). The third one is the doubly-robust performance estimator (equation 7). Taking these into consideration, let the combined performance function be, $\tilde{J}_{\mathrm{DR}}^{\lambda,\nu}(G_1, G_2) := J_{\mathrm{DR}}(\alpha_\pi^\nu(G_1), \alpha_w(\alpha_\pi^\nu(G_1), G_2), \alpha_f^\lambda(\alpha_w(\alpha_\pi^\nu(G_1), G_2), G_2), G_2), G_2)$, and the combined performance estimator be $\tilde{J}_{\mathrm{DR}}^{\lambda,\nu}(\hat{G}, \hat{G})$. We call the importance weight and the predictor an *evaluator*. Note that proposition 2 holds for the combined performance estimator by further assuming that $w$ is normalized and entire. Proposition 3 holds for the importance sampling performance estimator by further assuming the unbiasedness of the importance weight, but we have not found natural assumptions for the doubly-robust one. See appendix E for details.

### 3.2 REDUCING REUSING BIAS

Given the discussion in the previous section, let us define the reusing bias for any $\tilde{J} \in \{\tilde{J}_{\mathrm{PI}}, \tilde{J}_{\mathrm{IS}}, \tilde{J}_{\mathrm{DR}}, \tilde{J}_{\mathrm{DR}}^{\lambda,\nu}\}$ as, $b^N(G) := \mathbb{E}_{\hat{G} \sim G^N}\left[\tilde{J}(\hat{G}, \hat{G}) - \tilde{J}(\hat{G}, G)\right]$, and let us discuss how to reduce the reusing bias. Our approach is to estimate the reusing bias and substract it from the performance estimator. Such a bias reduction has been extensively discussed in the literature of information criteria (Konishi & Kitagawa, 2007), which aim to estimate the test performance of a predictor in a supervised learning setting by correcting the bias of its training performance. There are mainly two approaches: train-test split method and bootstrap method.

### 3.2.1 BIAS ESTIMATION BY TRAIN-TEST SPLIT

The first approach estimates the bias via train-test split of the sample. The sample $\mathcal{D}$ is randomly split into $\mathcal{D}_{\mathrm{train}}$ and $\mathcal{D}_{\mathrm{test}}$ such that $\mathcal{D}_{\mathrm{train}} \cap \mathcal{D}_{\mathrm{test}} = \emptyset$ and $\mathcal{D}_{\mathrm{train}} \cup \mathcal{D}_{\mathrm{test}} = \mathcal{D}$. Let $\hat{G}_{\mathrm{train}}$ and $\hat{G}_{\mathrm{test}}$ denote the corresponding empirical distributions. The reusing bias is estimated by $b_{\mathrm{split}}(\hat{G}) = \mathbb{E}[\tilde{J}(\hat{G}_{\mathrm{train}}, \hat{G}_{\mathrm{train}}) - J(\hat{G}_{\mathrm{train}}, \hat{G}_{\mathrm{test}})]$, where the expectation is with respect to the random split.

While this estimator seems to be reasonable, it is not recommended for our problem setting due to the bias of the bias estimator. As demonstrated in proposition 4, the train-test split estimator has $O(1/N)$ bias, the same order as the bias $b^N(G)$ itself, and therefore, it is not reliable. Such a bias is due to the non-linearlity of $\tilde{J}(G_1, G_2)$ with respect to $G_2$, the distribution used for testing[3]. See appendix D.2 for its proof and appendix G for the comparison with supervised learning.

**Proposition 4.** *Suppose we randomly divide the sample such that $|\mathcal{D}_{\mathrm{train}}| : |\mathcal{D}_{\mathrm{test}}| = \lambda : (1-\lambda)$ for some $\lambda \in (0,1)$. Under assumptions 10 and 12, $\mathbb{E}_{\hat{G} \sim G^N}[b_{\mathrm{split}}(\hat{G})] = b^N(G) + O(1/N)$ holds.*

Note that direct estimation of test performance by $\tilde{J}(\hat{G}_{\mathrm{train}}, \hat{G}_{\mathrm{test}})$ is not recommended similarly, unless the size of the test sample is sufficiently large. See appendix G for detailed discussion.

### 3.2.2 BOOTSTRAP BIAS ESTIMATION

An alternative approach to estimating the reusing bias (equation 4) is bootstrap (Efron & Tibshirani, 1994). A bootstrap estimator of the reusing bias $b^N(G)$ is obtained by plugging $\hat{G}$ into $G$: $b^N(\hat{G}) = \mathbb{E}_{\hat{G}^\star \sim \hat{G}^N}[\tilde{J}(\hat{G}^\star, \hat{G}^\star) - \tilde{J}(\hat{G}^\star, \hat{G})]$. Let $\hat{G}^{(m)}$ ($m \in [M]$) be a bootstrap sample obtained by uniform-randomly sampling data points $N$ times from the original sample $\hat{G}$ with replacement. Then, its Monte-Carlo approximation is, $\hat{b}^N(\hat{G}) = \frac{1}{M}\sum_{m=1}^{M}[\tilde{J}(\hat{G}^{(m)}, \hat{G}^{(m)}) - \tilde{J}(\hat{G}^{(m)}, \hat{G})]$. In contrast to

---

[3]The standard supervised learning scenario does not suffer from this bias because the performance estimator is linear with respect to the testing distribution.

the train-test split method, the bootstrap bias estimation can estimate the bias as stated in proposition 5. See appendix D.2 for its proof.

**Proposition 5.** *Under assumptions 10 and 12, $\mathbb{E}_{\hat{G} \sim G^N}[b^N(\hat{G})] = b^N(G) + O(1/N^2)$ holds.*

### 3.2.3 SUMMARY

We have introduced two reusing-bias estimators, referring to the literature of information criteria. We have found that the train-test split estimator, one of the most popular estimators, cannot reliably estimate the bias in our problem setting, although it works in supervised learning. In contrast, the bootstrap bias estimator is shown to be less biased than the train-test split estimator and can estimate the reusing bias more reliably. Therefore, we conclude that the bootstrap bias estimator is preferable to the train-test split estimator.

From computational point of view, the bootstrap bias estimator requires us to train $M$ agents and $M + 1$ evaluators. We set $M = 20$ in the experiments given the result of a preliminary experiment. Since the bootstrap procedure can be easily parallelized with low overhead, its wall-clock time can be reduced in proportion to the computational resource.

## 4 EMPIRICAL STUDIES

Let us empirically quantify the two biases as well as the effectiveness of the bias reduction methods. We first describe our experimental setup. See appendix H for full details to ensure reproduciability.

**Molecular representation.** All of the functions defined over molecules use the 1024-bit Morgan fingerprint (Morgan, 1965; Rogers & Hahn, 2010) with radius 2 as a feature extractor.

**Environment and Agent.** We employ the environment and the agent by Gottipati et al. (2020) with minor modifications. The agent receives a molecule as the current state, and outputs an action consisting of a reaction template and a reactant. The environment, receiving the action, applies the chemical reaction defined by the action to the current molecule to generate a product, which is then set as the next state. This procedure is repeated for $H$ times, and lastly the agent takes action $a_\perp$ to be rewarded by the property of the final product. We set $H = 1$ to reduce the variance in the estimated performance and better highlight the biases and their reduction. The agent is implemented by actor-critic using fully-connected neural networks.

We use the reaction templates curated by Button et al. (2019) and prepare the reactants from the set of commercially available substances in the same way as the original environment. The number of reaction templates is 64, 15 of which require one reactant, and 49 of which require two reactants. The number of reactants is 150,560.

**Evaluators.** As a predictor, we use a fully-connected neural network with one hidden layer of 96 units with softplus activations except for the last layer. It is trained by minimizing the risk defined over $S \sim G_S$. As the importance weight, we use the kernel unconstrained least-squares importance fitting (KuLSI) (Kanamori et al., 2012). In particular, we use the trained predictor except for the last linear transformation as a feature extractor and compute the linear kernel using it.

**Evaluation framework.** To evaluate the biases, we need the true property function $f^\star$, which however is not available in general. We thus design a semi-synthetic experiment using a real-world dataset $\mathcal{D}_0 = \{(m_n, f^\star(m_n)) \in \mathcal{S}^\star \times \mathbb{R}\}_{n=1}^{N_0}$. While any function $\mathcal{S}^\star \to \mathbb{R}$ can be used as the true property function $f^\star$, we substituted the predictor provided by Gottipati et al. (2020) for $f^\star$, which was trained with the ChEMBL database (Gaulton et al., 2017) to predict $\text{pIC}_{50}$ value associated with C-C chemokine receptor type 5 (CCR5). With this property function, we have full access to the environment, and we can construct an offline dataset $\mathcal{D}$ of an arbitrary sample size by running a random policy on $\mathcal{M}$, which is available in our setting.

To decompose the bias into the misspecification bias and the reusing bias, we need $f^\infty$, the predictor obtained with full access to the data-generating distribution $G$. We approximate it by $\alpha_f(\hat{G}_{\text{test}})$, where $\hat{G}_{\text{test}}$ is the empirical distribution induced by a large sample $\mathcal{D}_{\text{test}}$ of size $10^5$ constructed independently of $\mathcal{D}$. This approximation is valid if $|\mathcal{D}_{\text{test}}|$ is sufficiently large (see proposition 23). Then, the misspecification bias can be estimated by $\tilde{J}(\hat{G}, \hat{G}_{\text{test}}) - J^\star(\hat{\pi})$ and the reusing bias by

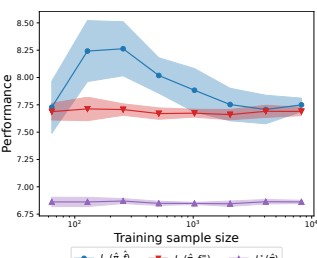 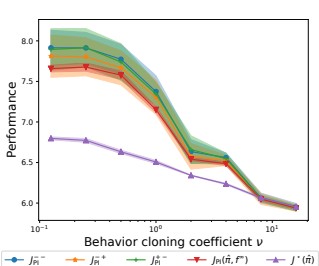 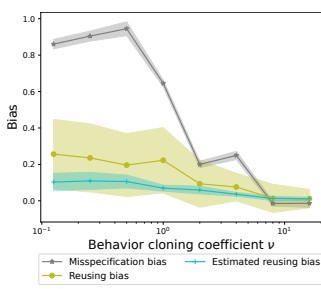

Figure 1: Lines show means and shaded areas show standard deviations. **(Left)** Biases vs. the sample size. $J_{\text{PI}}(\hat{\pi}, \hat{f}) - J_{\text{PI}}(\hat{\pi}, f^{\infty})$ corresponds to the reusing bias and $J_{\text{PI}}(\hat{\pi}, f^{\infty}) - J^{\star}(\hat{\pi})$ to the misspecification bias. **(Middle)** Comparison between bias reduction methods. **(Right)** Comparison between the misspecification bias, reusing bias, and the estimated reusing bias.

$\tilde{J}(\hat{G}, \hat{G}) - \tilde{J}(\hat{G}, \hat{G}_{\text{test}})$. The performance estimators are defined by the expectation with respect to a trajectory of a policy, and we estimate them by Monte-Carlo approximation with 1,000 trajectories.

**Quantifying the two biases.** First, we quantify the misspecification and reusing biases. In specific, we aim to study the relationship between these biases and the sample size. We vary the training sample size $N$ in $\{2^6, 2^7, \ldots, 2^{13}\}$. For each $N$, we generate five pairs of train and test sets, and evaluate the biases as indicated above. We report the means and standard deviations.

Figure 1 (left) illustrates the result. We have three observations. First, when $N = 2^7$, the misspecification bias, $J_{\text{PI}}(\hat{\pi}, f^{\infty}) - J^{\star}(\hat{\pi})$, was roughly twice as large as the reusing bias, $J_{\text{PI}}(\hat{\pi}, \hat{f}) - J_{\text{PI}}(\hat{\pi}, f^{\infty})$, demonstrating that both are non-negligible. Second, for $N \geq 2^7$, the reusing bias increased as the size of the training sample decreased, which coincides with proposition 2. The results for $N < 2^7$ did not coincide with it because the sample size is not large enough for asymptotic expansion to be justified. Third, the ground-truth performance of the policies was rather stable across different training sample sizes. We found that the policies were similar to each other, suggesting that this environment has a local optimum with a reasonably good performance (*cf.*, the performance of a random policy is around 5.8). This also suggests that the policy learner in our experiment was insensitive to the particular sample, and the reusing bias in this case is mainly caused by the finiteness of the sample to train the predictor, not by reusing the same sample.

**Quantifying Bias Reduction Methods.** We then study the effectiveness of the bias reduction methods presented in section 3. Since the behavior cloning coefficient $\nu$ will control the trade-off between the misspecification bias and the performance of the learned policy, it should be determined according to the user's requirement, *i.e.*, whether the accuracy of performance estimation or the actual performance is prioritized. Therefore, we design an experiment to evaluate the effectiveness of the bias reduction methods, varying $\nu$ in the range of $\{2^{-4}, \ldots, 2^4\}$.

Let $J_{\text{PI}}^{b_1 b_2}$ ($b_1, b_2 \in \{+, -\}$) be the plug-in performance estimator with covariate shift ($b_1 = +$) or without it ($b_1 = -$) and with bootstrap bias reduction ($b_2 = +$) or without it ($b_2 = -$). Let us define $J_{\text{DR}}^{b_1 b_2}$ accordingly for the doubly-robust performance estimator. We compare the performance estimates by $J_{\text{PI}}^{--}$, $J_{\text{PI}}^{+-}$, $J_{\text{PI}}^{-+}$, and $J_{\text{DR}}^{--}$ to see the effectiveness of each bias reduction strategy.

Figure 1 (middle) illustrates the performance estimates for $N = 10^3$. Since $J_{\text{DR}}^{--}$ performs significantly worse than the baseline $J_{\text{PI}}^{--}$, we omit it from the figure. See appendix I for the full result. We observe that the bootstrap bias reduction worked well, while the benefit of the covariate shift strategy is marginal. This indicates that the importance weight estimation did not work well in this setting.

Figure 1 (right) illustrates the biases in $J_{\text{PI}}^{--}$ and the reusing bias estimated by the bootstrap method. As we expected, the misspecification bias tends to decrease as we increase $\nu$. The reusing bias is under-estimated, but the estimated reusing bias contributes to bias correction.

In summary, we confirm that (i) behavior cloning can reduce the misspecification bias at the expense of performance degradation, (ii) the reusing bias can be estimated and corrected by bootstrap, and (iii) the methods using importance weights did not perform well in our setting.

## 5 RELATED WORK

Our primary contribution is the comprehensive study of theoretically-sound evaluation methodology for *in silico* molecular optimization algorithms using real-world data. Since the pioneering work by Gómez-Bombarelli et al. (2018), a number of studies on this topic have been published in the communities of machine learning and cheminformatics to advance the state-of-the-art. While some of them (Gómez-Bombarelli et al., 2016; Takeda et al., 2020; Das et al., 2021) have been validated *in vitro*, many others have been evaluated *in silico*.

Early studies (Kusner et al., 2017) adopted the octanol-water partition coefficient, $\log P$, penalized by the synthetic accessibility score (Ertl & Schuffenhauer, 2009) and the number of long rings as the target property to be maximized. The score can be easily computed by RDKit, and is often implicitly regarded as a reliable score computed by an accurate simulator. Some recently consider that the $\log P$ optimization is not appropriate as a benchmark task because it is easy to optimize (Brown et al., 2019) or its prediction can be inaccurate (Yang et al., 2021), and alternative benchmark tasks have been investigated; some of them propose a suite of benchmark tasks (Brown et al., 2019; Polykovskiy et al., 2020) and the others use other property functions trained by real-world data (Olivecrona et al., 2017; Li et al., 2018a; Jin et al., 2020; Gottipati et al., 2020; Xie et al., 2021). However, most of the current evaluation protocols rely on the naive plug-in performance estimator.

As far as we are aware of, there are at least two empirical studies concerning about potential biases in the plug-in performance estimator. Renz et al. (2019) pointed out that the plug-in performance estimator is biased due to data reuse and random initialization of the predictor, while a follow-up study by Langevin et al. (2022) attributed the bias to the train-test split used by Renz et al. (2019); the train and test sets were far from being identically distributed. While these two pioneering studies shed light on the potential flaw in the plug-in performance estimator, we have not fully understand it partially because these studies are empirical.

Our contribution to this line of studies is that we not only empirically but also theoretically demonstrate potential biases in the current evaluation methodology and present bias reduction methods. This also unveils why the $\log P$ optimization task has been hacked and suggests that the alternative benchmark tasks will be hacked as long as no bias reduction method is applied. The $\log P$ function implemented in RDKit (Wildman & Crippen, 1999) is obtained by fitting a linear model to a dataset of experimental $\log P$ values, and is in fact a predictor. Our theory suggests that unless the bias reduction methods are applied, the learned agent generates unrealistic molecules that are far from those in the dataset (which has been often reported in $\log P$ optimization), and the resultant performance estimate is biased. This mechanism is also valid for the alternative benchmark tasks, and we conjecture they will also be hacked sooner or later. It also suggests that by incorporating bias reduction methods, we can reliably estimate the performance and therefore can safely compare different methods even when using the $\log P$ optimization task.

Our work shares a similar objective with a seminal work by Ito et al. (2018), which aims to reduce the reusing bias that appears when solving an optimization problem whose parameters are estimated from data. A major contribution to this literature is to relax their assummption that the predictor is well-specified. This introduces the concept of misspecification bias, which was confirmed to be non-negligible in our application. Another minor contribution is to formalize their reusing-bias correction method by bootstrap and investigate the theoretical properties.

## 6 CONCLUSION AND FUTURE WORK

We have discussed that the plug-in performance estimator is biased in two ways; one is due to model misspecification and the other is due to reusing the same dataset for training and testing. In order to reduce these biases to obtain more accurate estimates, we recommend to (i) add a constraint to the policy such that the state distribution stays close to the data distribution and (ii) correct the bias by bootstrapping if it is non-negligible and we can afford to do it.

A future research direction is to improve the importance weight estimation so that the other bias reduction methods work. Another is to constrain a policy with less performance degradation. Since the methods using variational autoencoders (Gómez-Bombarelli et al., 2018; Jin et al., 2018; Kajino, 2019) can naturally generate molecules similar to those in the data, such methods could be reevaluated.

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
