# OpenReview forum: "Biases in Evaluation of Molecular Optimization Methods and Bias Reduction Strategies"
_ICLR.cc/2023/Conference — Submitted to ICLR 2023_

### Official Review · Reviewer_rwC3 · 2022-10-24

**Confidence:** 4
**Correctness:** 3
**Technical Novelty And Significance:** 2
**Empirical Novelty And Significance:** Not applicable
**Recommendation:** 3

**Clarity, Quality, Novelty And Reproducibility:**

The paper is well written with sufficient supplementary information, and the issues and problems analyzed in the paper are definitely important and interesting, and presumably reproducible ones. In contrast, novelty wise, I am still unsure, as I described several concerns in the above [Weakness] section.

**Strength And Weaknesses:**

[Strength]

- This paper focuses on very interesting and important bias problems when we use ML for molecular optimization or conditioned molecular generation for optimizing a given target property. A common practice would be the "direct method or plug-in method," that first learns a model of the system and then use it to estimate the performance of the evaluation policy, which causes misspecification bias and reusing bias.

- Theoretical considerations are grounded on a general setup of Markov decision process with some mild conditions.

- Not only providing theoretical results, but the paper also provides actionable practical solutions to reduce the analyzed biases. In particular, the uses of (i) importance sampling estimators, (ii) doubly robust estimators, and (iii) bootstrapping/cross-validation.


[Weakness]

- As in the paper's title, this paper targets molecular optimization, but the theoretical/technical considerations were done in a very general and abstract setup, and the obtained implications from theoretical parts were unclear. This paper's theoretical part might be better to be reconsidered in a much broader setup. For examples of three main results (Theorem1, Proposition 2, Proposition 3), Theorem 1 seemed a general decomposition, Proposition 2 is an accurate estimate of the bias WITHOUT comparing to other unavoidable uncertainty (at least, need to relate this to some uncertainty quantification of aleatoric and epistemic uncertainties?), Proposition 3 just provides positivity not referring whether this amount is negligible or not. So from the theoretical parts, we are still not sure if these biases are problematic or negligible.

- The fact that theoretical analyses are mathematically solid negatively emphasizes the gap between practical situations and formal definitions. For example, the paper formulated molecular optimization as reinforcement learning to maximize the "expected cumulative reward," but the goal of molecular optimization is usually just to have better-performing molecules than originals and doesn't need to maximize "cumulative" scores or doesn't have nice intermediate molecules during optimization. In this sense, the target task sounds like "pure exploration" rather than RL having an exploitation-exploration dilemma. "maximum reward" (for example, below [1][2]) would be more natural than "cumulative or averaged reward," but this change seems to violate some basic assumptions on continuity of this paper. (For example, the entirety of Banach space?), and shake the values of the obtained theoretical results.

[1] Quah & Quek, Maximum reward reinforcement learning: A non-cumulative reward criterion. (2006)
    https://doi.org/10.1016/j.eswa.2005.09.054

[2] Gottipati et al., Maximum Reward Formulation In Reinforcement Learning. (2020)
    https://arxiv.org/abs/2010.03744

- The entire contents share strong similarity to the similar discussion in OPE (off-policy evaluation), and would be incremental given that  OPE is one of the RL-related recent hot topics in ML. The paper cited DudÍk et al. 2014 in a contextual bandit context, but this work came from ICML 2011 by the same authors https://arxiv.org/abs/1103.4601 and relevant research along this line is now being intensively investigated in ML. One of the highly cited results in this context would be [3] below, and this paper analyzed the plugin bias under study as "DM (direct method)," "IS (importance sampling method)" such as Inverse Probability Weighting (IPW), and "DR (doubly robust)," which is very similar to this paper. Of course, full RL (in this paper) and contextual bandits/OPE would differ in many points, but these relevant contexts should be incorporated to make contributions much clearer.

[3] Farajtabar et al, More Robust Doubly Robust Off-policy Evaluation. (ICML 2018)
    https://arxiv.org/abs/1802.03493

- Similarly, the use of some out-of-sample estimators (like cross-validation or bootstrapping) sounds like standard practice for these situations (plug-in situations?), and I'm not that clear that this proposal is somewhat novel. At least, the use of bootstrapping in OPE (for example [4] below) also needs to be appropriately positioned in a related context.

[4] Hao et al., Bootstrapping Fitted Q-Evaluation for Off-Policy Inference (ICML 2021)
    https://arxiv.org/abs/2102.03607

**Summary Of The Paper:**

This paper targets molecular optimization, where we want to modify/generate the structure of molecules to optimize some target properties. For this problem, this paper provided two contributions: 1) theoretical clarification on problematic biases for a common practice of using the same data for both training and evaluating the generator. 2) practical solutions to these biases, particularly the uses of (i) importance sampling estimators, (ii) doubly robust estimators, and (iii) bootstrapping/cross-validation. Empirical studies were also made using the setup of molecular optimization as reinforcement learning by Gottipati et al. (2000).

**Summary Of The Review:**

This paper studied important bias problems in RL as a context in molecular optimization. However, it has unclear implications and contexts: 1) why this paper needs to target molecular optimization is unconvincing because both theoretical and practical parts seemed to be discussed in a much more general and abstract setup, and implications of the obtained theorems in the context of molecular optimization are unclear. 2) The use of "cumulative" reward sounds a bit unnatural for the goal of molecular optimization (Though this might just come from the other existing work...). 3) The contents of practice (bias reduction) parts are similar to well-known discussions in an OPE or contextual bandit context. In particularly, "DM(direct method)" corresponds to "the plugin bias" in this paper, "IS(importance sampling method)" corresponds to "covariate shift" in this paper, "DR(doubly robust)" corresponds to DR (3.1.3) in this paper.

---

> ### Author Response · Authors · 2022-11-11
> **We would like to discuss the relationship to offline RL.**
>
> We would like to thank Reviewer rwC3 for detailed comments and insightful suggestions. Below are responses specific to Reviewer rwC3, and also please refer to the general responses.
>
> __Why targeting molecular optimization:__ This is related to both our initial motivation and strategy to make greater impact. We have studied molecular optimization methods to find that the performance evaluation seems to be incorrect. To find out the cause of it, we theoretically analyze the performance estimator, which results in the research presented in the current paper. As pointed out by Reviewer rwC3, the analysis presented in the paper and the methods may be generalized to offline RL; however, we did not do that because molecular optimization is more applied to real-world problems than offline RL and there is great urgency to report the common pitfall that many of existing studies have overlooked; otherwise the molecular optimization community could head to wrong directions. In order to make the greatest impact, we chose to focus on the molecular optimization domain, instead of generalizing it to offline RL.
>
> __Implications from theoretical parts:__ Proposition 2 implies that the reusing bias is O(1/N), and Equation 3 implies that the misspecification is likely to be $O(1)$ (because the misspecification bias is inherent to the misspecification of the function class of the predictor). They show that if the sample size is small, both of the biases could be present, and even if the sample size is large enough, the misspecification never vanishes.
>
> __Our MDP formulation:__ Although our formulation uses the cumulative reward, since the agent is rewarded only once when it terminates (it can terminate anytime when it wishes) and outputs a molecule that the agent considers optimal; the agent is not necessarily encouraged to obtain better intermediate molecules, but is encouraged to output one optimal molecule that the agent considers optimal. Therefore, even without relying on the maximum reward formulation, our formulation encodes the user’s objective to find optimal molecules. In addition, the entirety assumption is used to quantify the reusing bias, and other results including the bias decomposition, characterization of the misspecification bias, and the policy constraining approach to reduce it remain true.
>
> __Relation to offline RL:__ We are grateful for pointing it out. A unique attribute of our research against offline RL is that we unify OPE and offline policy optimization, meaning that we learn a policy $\hat{\pi}$ from a dataset and at the same time we evaluate its policy using the same dataset. In contrast, many of OPE methods focus on evaluating a given policy which is not dependent on a dataset for OPE. This formulation allows us to develop our bias decomposition (Theorem 1), because the reusing bias is inherent to our formulation. In addition, this attribute allows us to come up with the policy constraining approach to reduce the misspecification bias, where the intuition is to optimize the policy within the region where policy evaluation is reliable.
>
> __Relation to offline RL 2:__ We would like to discuss the relationship between our bias reduction methods and the methods in offline RL.
> - DM in offline RL = plug-in estimator in our paper
> - IS in offline RL = importance sampling estimator in our paper (Section 3.1.3)
> - Doubly robust estimator in offline RL = doubly robust performance estimator in our paper (Section 3.1.3).
>
> But we are not aware of any correspondence in offline RL to the covariate shift explained in Section 3.1.1; in offline RL, its idea can be translated into offline RL such that it learns the environment by considering the distribution shift between the data distribution and the distribution induced by the policy to be evaluated. If there exists any, we would appreciate it if Reviewer rwC3 could point it out.
>
> We consider that the methods to reduce the bias are not very novel, because they are adapted from existing literatures; but we consider that it is valuable to summarize existing approaches that can be used to reduce the two biases that we defined.

---

> > ### Comment · Reviewer_rwC3 · 2022-11-22
> > **Thanks for the clarification, and summary of my thoughts**
> >
> > Thank you for the clarification, and sorry for this late post. Seems like I made a mistake on the discussion schedule...
> >
> > My evaluation was almost the same as the Official Review by Reviewer zS58. The difference in rating comes from my impression that the paper would be in a better form if it is reorganized in a much broader setup by also considering the strongly related topics such as "Off-policy Evaluation (OPE)" or "Off-policy Learning (OPL)" context. In my option, the paper's interest is highly overlapped with the interests of OPE or OPL in the sense that we are interested in the gap between the data at hand and what we truly want for deciding our action.
> >
> > Also, I'm still not convinced about the repeatedly-claimed point "many use generators trained in data X, and evaluate it by also using the data X used for training." The molecular optimization community didn't practice any cross-validation...!? What if we just split the dataset into train and test, train the generator on the train, and evaluate it using test...?

---

> > > ### Author Response · Authors · 2022-11-22
> > > **Two major points to be discussed**
> > >
> > > Thank you very much for further responding to our responses and for the insightful suggestions. It seems that we are not agreeing on,
> > > 1. whether the data reuse is the case or not, and
> > > 2. whether our paper should target at offline RL or molecular optimization,
> > >
> > > and let us present our positions on them. We hope that they are helpful to reach a consensus between us. If you have any other points to be discussed, please let us know.
> > >
> > > __Reusing data for training and testing:__
> > > This data reuse can occur unintentionally when we substitute a predictor $\\hat{f}$ for the oracle $f^\\star$; we typically use $\\hat{f}$ to train the policy *and* evaluate the policy.
> > > We have listed several papers that reuse the same predictor for training and testing: https://openreview.net/forum?id=Sh97TNO5YY_&noteId=GDsTVDZu0W2
> > > In addition, the famous penalized log P benchmark also reuses the predictor, because `Descriptors.MolLogP` in `rdkit` is a predictor.
> > > Therefore, this is the case.
> > >
> > > Additionally, even if we split the dataset into train and test sets and obtain two predictors, $\\hat{f}_\\mathrm{train}$ and $\\hat{f}_\\mathrm{test}$, the resulting performance evaluation is still biased in the same way as the original analysis, as elaborated in
> > >
> > > https://openreview.net/forum?id=Sh97TNO5YY_&noteId=_tQmdDiBKe0
> > >
> > > That is, the bias can be still decomposed into two terms, one is the misspecification bias, and the other is the *finiteness bias*, which is due to the finiteness of the dataset used to train $\\hat{f}_\\mathrm{test}$. In other words, the reusing bias in the original manuscript is caused by both data reuse and the finiteness of the data, as suggested in footnote 1, page 3, and even if we employ the train-test split strategy, the finiteness bias remains.
> > > Therefore, the train-test split does not completely address the bias issue reported in this paper, and there remain both the finiteness bias and the misspecification bias.
> > >
> > > __Appeal to the offline RL community:__
> > > After receiving the review by Reviewer rwC3, we have wondered about this possibility. However, we consider there is a non-trivial gap between the current formulation and offline RL settings, and we still consider the paper should target at the molecular optimization community, rather than in the offline RL community. Let us explain the gap we have noticed.
> > >
> > > In our formulation, we assume that,
> > > - the state dynamics of the MDP, $p(s^\\prime \\mid s, a)$, is known
> > > - the reward functions at steps $0,1,\\dots,H-1$ are known, and
> > > - the reward function at step $H$ is the only unknown component.
> > >
> > > In contrast, a standard offline RL assumes that all of them are unknown.
> > >
> > > Our theoretical analyses are still applicable to the case when all of the reward functions are unknown, but are *not* directly applicable to the case when the state dynamics is unknown. In particular, the characterization of the misspecification bias (Eq. 3) does not hold, which is essential to derive a method to reduce the misspecification bias.
> > > In other words, our results rely on the assumption that the state dynamics is known, which is inherent to molecular optimization while is not natural for general offline RL settings.
> > > Therefore, we consider the present work, which we believe has enough contributions to be published as one research paper, should target at molecular optimization, rather than offline RL (of course, we will discuss the relationship to it with appropriate citations).

---

> > > > ### Comment · Reviewer_rwC3 · 2022-11-24
> > > > **My views on these two points**
> > > >
> > > > Thanks for the clear statements, and I think I understand the point. Here I reword my views on these points.
> > > >
> > > > > 2. whether our paper should target at offline RL or molecular optimization,
> > > >
> > > > This point was more important to me, and it is very informative to see the non-trivial gap pointed out here. So the "molecular optimization" setup would be a specialized setup of the general offline RL that assumes several fundamental quantities known we usually set unknown in RL (in particular, the paper assumes that the state dynamics are all known).
> > > >
> > > > But as Reviewer zS58 and I pointed out in the Official Review, the standard MDP setup would be a bit unnatural starting point if the paper claims that molecular **optimization** is the real target even though some existing work (impractically?) used it. We will prefer a simpler **optimization** setup because the goal of molecular optimization should be to find any highly optimized molecules rather than to maximize the expected cumulative quantities. That's why I thought it is better to treat molecular optimization as a use-case example and discuss the entire results in a much broader setup with presented unusual 'known' assumptions in RL.
> > > >
> > > > > 1. whether the data reuse is the case or not, and
> > > >
> > > > It was also informative to see the point about misspecification bias and the finiteness bias, but also I feel this 'reuse' might be included as a special case in the main interest of off-policy evaluation (OPE) and off-policy learning (OPL). So potential readers would like to understand how the presented results of this paper and the established results in OPE/OPL are interrelated.
> > > >
> > > > At least, the presented solutions are highly overlapped as follows, plus "doubly-robust" estimators are borrowed from OPE/OPL research, and I think the paper should not ignore this context.
> > > >
> > > > Ensure that "the goal (of OPE in RL) is to estimate the performance of a policy from the data generated by another policy(ies) [2]". So this would also be the case for general model-based optimization or model-based RL when we use any ML surrogate for the oracle but want to extend its prediction out of the training set for the ML surrogate (or even counterfactual situations that the surrogate never sees in training).
> > > >
> > > > - Direct Method (DM) in [1][2] = plug-in estimator in the paper
> > > > - Importance Sampling (IS) or Inverse Propensity Score (IPS) in [1][2] = importance sampling estimator in the paper (Section 3.1.3)
> > > > - Doubly robust estimator in [1][2] = doubly robust performance estimator in the paper (Section 3.1.3).
> > > >
> > > > [1] Dudik, Langford, Li, Doubly Robust Policy Evaluation and Learning (ICML 2011) https://arxiv.org/abs/1103.4601
> > > > [2] Farajtabar et al, More Robust Doubly Robust Off-policy Evaluation. (ICML 2018) https://arxiv.org/abs/1802.03493
> > > >
> > > >
> > > > ## Further thought
> > > >
> > > > To be clear, if we accept the gap in the objective (or if we forget about molecular optimization?), I found the presented results very interesting. As Reviewer zS58 also mentioned, the basic decomposition is the bias-variance tradeoff also in OPE. For example, Proposition 1 of the paper [2] above is this one for DM (direct method = the "plug-in" version of this paper), and we can see that 'misspecification bias' is indeed unavoidable if the ML surrogate is biased (though the paper also considers an importance-sampling part, corresponding to the 'reuse' bias in this paper's setup).
> > > >
> > > > The estimator correction in the OPE context considers the variance (as Theore 1 and 2 in the case of [2]), and it would not the same as the paper's interest, but I thought the presented "misspecification bias"or 'reuse bias' (or 'finiteness bias'?) is closely related to these established discussions in OPE/OPL, and it would be very nice to see much clearer descriptions and discussion. (At least, better than presenting asymptotic results with less clear practical implications for molecular optimization...?)

---

> > > > > ### Author Response · Authors · 2022-11-24
> > > > > **Relationship to the existing OPE literature**
> > > > >
> > > > > Thank you very much for engaging yourself in this discussion. In this post, let us clarify the relationship to the existing OPE literature, in specific, the doubly robust OPE studies suggested by Reveiwer rwC3.
> > > > >
> > > > > As Reviewer rwC3 pointed out, the doubly robust estimators are related to our study in that we used its idea to one of our solutions (Section 3.1.3), and actually, we have cited the journal version of the research by Dudik et al. (2011). In order to compare our results with the existing literature, let us first assume that the policy to be evaluated is fixed (though our formulation assumes that the policy is dependent on data).
> > > > >
> > > > > Then, the bias of our interest is identical to the bias the existing studies have investigated (e.g., the bias in Proposition 1 in the paper by Farajtabar et al. (ICML18)). The main difference between them is that we decompose the bias into the stochastic term (i.e., finiteness bias, since the policy is assumed to be fixed in this discussion) and the deterministic term (the misspecification bias), whereas the existing studies did not. This difference leads to different conditions for the estimators to be unbiased.
> > > > > - Implications from the __existing__ study: use an unbiased estimator
> > > > >   - It concludes that the estimator of the reward function $\\hat{f}$ has to be unbiased even in the __finite__ sample situation, i.e., $\\mathbb{E}\\hat{f} = f^\\star$.
> > > > > - Implications from __our__ study: use an __asymptotically__ unbiased estimator + correct the finiteness bias by bootstrap
> > > > >   - $\\hat{f}$ has to be unbiased only in the __infinite__ sample situation (this immediately follows by setting the middle term of Eq. 3 equal to zero, i.e., $f^\\infty=f^\\star$), which is a weaker assumption than $\\mathbb{E}\\hat{f}=f^\\star$, and
> > > > >   - the finiteness bias can be corrected by estimating it by bootstrap methods or other statistical methods and by subtracting the estimated bias from the value.
> > > > >
> > > > > However, as elaborated in the previous post, since our study assumes that the state dynamics is known, our results are not trivially extended to the offline RL setting. For example, while we assume that $S_H$ is distributed according to the same distribution in Eq. 3, which will not be true when the state dynamics is unknown and has to be estimated.
> > > > >
> > > > > We would like to discuss this relationship in the related work, which will be informative to the offline RL community.

---

> > > > > > ### Comment · Reviewer_zS58 · 2022-11-28
> > > > > > **Thank you for this; I agree more discussion of OPE literature would help**
> > > > > >
> > > > > > I agree with reviewer rwC3's points that your work is very related to OPE and it would be good to see more discussion of this. I am not super familiar with this area of the literature, but I see similarities between the methods, and I agree with reviewer rwC3's remark that the results presented in this paper are applicable to much more than to molecules. Perhaps this means the scope of the paper could be broadened?

---

> > > > > > > ### Author Response · Authors · 2022-11-29
> > > > > > > **Relationship to OPE**
> > > > > > >
> > > > > > > We would like to thank Reviewer zS58 for suggestions. Since the relationships to OPE have been discussed in multiple posts, we would like to summarize them below.
> > > > > > >
> > > > > > > First of all, let us discuss why our results are not easily extended to a general OPE setting.
> > > > > > > - Our study assumes that most part of the environment except for the reward function at the final step is known, while both the state dynamics and the reward function are unknown in a standard OPE setting.
> > > > > > >   - Our setting allows us to derive a simple form of the misspecification bias (Eq. 3); otherwise, the expectation operator $\\mathbb{E}^\\hat{\\pi}$ will depend on estimated state dynamics, and the relationship to covariate shift will become unclear.
> > > > > > >   - The assumption that we know most of the environment is inherent to molecular optimization, and therefore, we consider not all of our results are easily extended to a general OPE setting.
> > > > > > > In addition, since we have clarified that our doubly-robust performance estimator is adopted from OPE (see the first paragraph of Section 3.1.3), we are not neglecting the OPE literature even in the original manuscript.
> > > > > > >
> > > > > > > Below are interesting relationships between our results and those in the OPE literature, which are worth discussing in the paper.
> > > > > > > - We handle offline policy evaluation and optimization at the same time, while many OPE studies assume that the policy to be evaluated is given. This difference allows us to come up with our policy constraining approach to reducing the misspecification bias (see "Relation to offline RL" in [this post](https://openreview.net/forum?id=Sh97TNO5YY_&noteId=h0kRb2WZfm)).
> > > > > > > - Even if the policy to be evaluated is fixed and given, our formulation is different from the existing one because of our bias decomposition. Existing studies in the OPE literature require us to use an unbiased estimator, while our analysis suggests that an *asymptotically* unbiased estimator is ok if we correct the finiteness bias by bootstrap or if the finiteness bias is small (see [this post](https://openreview.net/forum?id=Sh97TNO5YY_&noteId=cmBM7fgw5p)). An asymptotically unbiased estimator is much easier to develop, because it essentially requires that the model class contains the true function $f^\\star$.
> > > > > > >
> > > > > > > In summary, while there are interesting connections between our setting and the OPE setting,
> > > > > > > 1. our analysis retains some novelty and significance over the existing OPE analyses, and
> > > > > > > 2. a part of our analysis relies on the assumption that most of the environment is known, which is true for molecular optimization but not for general offline RL settings, and therefore, our result is not suitable to be presented as a general OPE paper, but is reasonably presented as a molecular optimization paper (with some discussion on the relationships to OPE as described above).

---

> > > > > ### Author Response · Authors · 2022-11-24
> > > > > **On expected reward in molecular optimization**
> > > > >
> > > > > We understand Reviewer rwC3's concern that the __expected__ reward may be inconsistent with the purpose of molecular optimization, which sometimes pursues the __maximum__ reward observed during generation. First of all, please notice that we have elaborated in the discussion with Reviewer zS58 that there are not a few studies that use the expectation-based score as the objective function, and our formulation is directly applicable to them. Then, let us discuss why we prefer the expectation-based formulation to the maximum-based formulation.
> > > > >
> > > > > There are two reasons. The first reason is that the expectation-based formulation matches the objective function of RL-based molecular optimization methods, and is easier to estimate and is easier to handle than the tail statistics (e.g., top-5 scores). Therefore, the expectation-based formulation will allow us to avoid unnecessarily difficult mathematics and to present the flaw of the current evaluation protocol more clearly (though, we are struggling with the presentation).
> > > > >
> > > > > The other reason is that we consider it is more consistent to real applications than the top-k scorings. As elaborated in the introduction of the Guacamol paper [1], one of the main applications of molecular optimization methods is to complement virtual screening. At the virtual screening stage of drug discovery, it is desirable to have as many molecules as possible that possess the desirable properties. This indicates that situation 1 below is better than situation 2:
> > > > > 1. the molecules sampled from the optimized generator are good in average,
> > > > > 2. the top-k (k~10 to 100) molecules sampled from the optimized generator are good
> > > > >
> > > > > In other words, for the virtual screening application, a generator that generates many good molecules should be preferred to a generator that generates a few very good molecules but many not-good molecules. To reflect this demand, we prefer the expectation-based objective function, instead of the maximum-based one (although not a few studies have not clarified the exact application of their methods and simply seek a few good molecules, which we consider another flaw of the current benchmark).
> > > > >
> > > > > We admit that it is very interesting to extend our work by replacing the expectation in the objective function with another operator that seeks the upper-tail of the distribution of $f(S_H)$. Since such an extension is non-trivial, we leave it for future work.
> > > > >
> > > > > - [1]: Brown et al., "GuacaMol: Benchmarking Models for De Novo Molecular Design"

---

> > > > > > ### Comment · Reviewer_rwC3 · 2022-11-24
> > > > > > **I respectfully disagree that " for the virtual screening application, a generator that generates many good molecules should be preferred to a generator that generates a few very good molecules but many not-good molecules. "**
> > > > > >
> > > > > > Thanks, I understand that the expectation-based formulation "avoid unnecessarily difficult mathematics", but respectfully disagree that " for the virtual screening application, a generator that generates many good molecules should be preferred to a generator that generates a few very good molecules but many not-good molecules. "
> > > > > >
> > > > > > The goal of virtual screening is to find promising initial compounds for drug development, and it is followed by subsequent phases as H2L (hit to lead) -> lead optimization (LO) -> preclinical development -> clinical development. The subsequent phases are costly in time and money, and thus I guess "a few very good molecules" would be more preferred than "many so-so molecules". Probably this is why people prefer top-k scoring. I believe 5 super-good molecules are better than 10000 mediocre-performing molecules. Since this is a search problem, it is even usually unknown whether we can have "many" such good molecules for the target, and only a surprisingly good molecule can become a drug finally. This is like looking for a needle in a haystack, and a large number of seemingly good molecules cannot be a real drug in usual.

---

> > > > > > > ### Author Response · Authors · 2022-11-24
> > > > > > > **An expected utility formulation can alleviate the issue**
> > > > > > >
> > > > > > > Thank you very much for your prompt response. We respect your position and we hope all the reviewers discuss whether the expectation-based formulation is ok or not.
> > > > > > >
> > > > > > > After consideration, we find that the expectation-based formulation can behave like the top-k score by transforming a reward by a utility function $u(r)$. For example, if we convert a reward from $r(s,a)$ to $\\exp(C r(s,a))$ for some constant $C>0$, the expected reward will prefer larger rewards to smaller ones, which realizes a score such that “5 super-good molecules are better than 10000 mediocre-performing molecules.” In the extreme case $C\\to\\infty$, the transformed score only sees the top-1 molecule. Therefore, we can conclude that our formulation can handle a score such that “5 super-good molecules are better than 10000 mediocre-performing molecules”, and such scores also suffer from the biases we have discussed.

---

> > > > > > > > ### Comment · Reviewer_zS58 · 2022-11-28
> > > > > > > > **Expectation is not a complete replacement for tail statistics**
> > > > > > > >
> > > > > > > > For example, consider the metric of the 99th percentile of generated molecules (i.e. reward based on top 1% of molecules). Although this formulation could result in the best molecules giving the most reward, expected value cannot distinguish between a distribution with a high top 1% and a distribution with a low top 1% but a _really_ high top 0.01% (for example). I think the effect would be similar but I do not think that expected value can be used to completely encompass metrics based on tail statistics.

---

> > > > > > > > > ### Author Response · Authors · 2022-11-28
> > > > > > > > > **Discussions on risk measures could be helpful**
> > > > > > > > >
> > > > > > > > > Thank you very much for suggesting to use the percentile as a metric. We consider that both the percentile metric and [our expected-utility metric](https://openreview.net/forum?id=Sh97TNO5YY_&noteId=rtCTryUNh6s) are *risk measures* that quantify the tail of a distribution, and therefore, our expected-utility metric is also acceptable (and both will be able to distinguish the examples Reviewer zS58 has provided).
> > > > > > > > >
> > > > > > > > > In fact, we observe a concurrent discussion in the domain of financial risk measures. The suggested percentile metric can be regarded as the value-at-risk in finance, whereas our expected-utility metric can be regarded as a variant of the entropic risk measure. Given both are regarded as risk measures quantifying the tail-behavior of a distribution, we consider both are acceptable metrics.

---

> > > > > > > ### Comment · Reviewer_zS58 · 2022-11-28
> > > > > > > **Agree with reviewer that expectation is not a reasonable metric**
> > > > > > >
> > > > > > > I agree with what the reviewer wrote, and did not find the arguments by the authors convincing.
> > > > > > >
> > > > > > > > The first reason is that the expectation-based formulation matches the objective function of RL-based molecular optimization methods, and is easier to estimate and is easier to handle than the tail statistics (e.g., top-5 scores).
> > > > > > >
> > > > > > > I don't think this makes much sense: it sounds like you are advocating for using metrics which are easy to estimate but not very relevant over metrics which are relevant. I agree tail statistics are hard to estimate but that doesn't mean researchers should use different metrics.
> > > > > > >
> > > > > > > > The other reason is that we consider it is more consistent to real applications than the top-k scorings. As elaborated in the introduction of the Guacamol paper [1], one of the main applications of molecular optimization methods is to complement virtual screening.
> > > > > > >
> > > > > > > Virtual screening is essentially just about tail statistics! The goal is to pick a small subset of molecules from a much larger database of molecules (i.e. find the tail of the distribution). If anything this seems like an argument to use tail statistics!

---

> > ### Comment · Reviewer_rwC3 · 2022-11-22
> > **Detailed comments for each point**
> >
> > > Why targeting molecular optimization:
> >
> > The intention is well taken. I agree that "there is great urgency to report the common pitfall that many of existing studies have overlooked;". But the other reviewer also pointed out, this problem is now already widely recognized among researchers, as seen in the GuacoMol paper, the paper in the comment by Reviewer zS58, and the other examples like [1]
> >
> > [1] Walters, W.P., Murcko, M. Assessing the impact of generative AI on medicinal chemistry. Nat Biotechnol 38, 143–145 (2020).
> >
> > So we should not be evaluated just by this urgency, and rather grounded by the paper's contribution itself.
> >
> > From this viewpoint, I felt that publishing the current result with placing a strong focus on "molecular optimization" is confusing and misleading from two reasons:
> >
> > (1) If molecular **optimization** is the target task, a straightforward approach would be model-based **optimization** like Bayesian optimization rather than RL maximizing "accumulated" reward. For the reuse bias, practitioners would carefully prepare the train/validation/test dataset, and would train the generator only using 'train' part to avoid any potential unintended data leakage. To me, paper's criticism would not apply to this typical approach in practice and I just wonder how much this practice is the case.
> >
> > Furthermore, as Reviewer zS58 also pointed out, the purpose of molecular optimization is to get maximally-performing molecules rather than to maximize the **cumulative** reward. So when a generator produces 100 candidates, it would be totally fine if we can find 1 very good molecule + 99 very poor molecules (at least, better than 100 mediocre molecules just similar to the given training data). If this is the case, the paper's starting point is a bit ill-calibrated, and any intensive theoretical analysis on it would be simply misleading or even potentially harmful if the paper's intention is to widely notify the problematic biases to the community (because this "setup gap" will induce a different type of unignorable biases...!?).
> >
> > The difference of the objective of (model-based) optimization and RL would be unignorable. Maximizing the cumulative reward requires solving the exploitation-exploration tradeoff, but molecular optimization here would be rather a pure exploration (pure search) problem to search any high-performing molecules that's why molecular exploration is primarily guided by model-based optimization like BO (or ay sequential experimental design).
> >
> > This would be very similar to the relationship between bandit and best-arm identification. These two are interrelated but optimal algorithm would be different. For this, I know we commonly observe that "ill-callibrated UCB-algorithms minimize regret while still identifying quickly the best arm [2]", and I guess that the presented method might work whether the theoretical setup is relevant or not. But this paper's contribution is theoretically grounded, and I thought we should not on this empirical fact...
> >
> > [2] Bridging the gap between regret minimization and best arm identification, with application to A/B tests (AISTATS 2019) https://arxiv.org/abs/1810.04088
> >
> > Remark: We might be able to think exploitation-exploration tradeoff when we think about any factors other than the target property, like financial costs, hardness of synthesis, availability, etc.
> >
> >
> > (2) I found the paper's theoretical contribution interesting in a much broader context but having this title focused on "molecular optimization" would not result in having proper reviewers in the related topics of offline RL, off-policy evaluation, off-policy learning, etc. This is sad if the paper's intention is to give practical impact to the community by notifying this problems of biases. So I thought this paper would be better to be reorganized in a broader setup and use "molecular optimization" as one of important use cases.
> >
> > To me, the paper's plot is too similar to the ones we see in OPE/OPL discussions [3] in the contextual bandit problem, which might be more relevant than full RL (for sequential decision making) for this paper's goal of molecular optimization (for single-shot decision making?).
> >
> > [3] Farajtabar et al, More Robust Doubly Robust Off-policy Evaluation. (ICML 2018) https://arxiv.org/abs/1802.03493

---

> > ### Comment · Reviewer_rwC3 · 2022-11-22
> > **Detailed point 2**
> >
> > > Implications from theoretical parts
> >
> > Yes, I found it interesting to see that the paper organized these points in a theoretically considerable setup, and come up with theorems and actionable solutions. That's why I thought that this paper would be better to be presented and discussed in much appropriate context (I'm not sure it's offline RL or contextual bandit though).
> >
> > At the same time, O(1) and O(1/N) are in an asymptotic sense, not exact bounds for convergence rates (a limiting behavior when N to infinity), and this weak theorems would not have any strong practical implications. We empirically know these biases are problematic, but I though theoretically this point would still be unclear.
> >
> >
> > > Our MDP formulation:
> >
> > This response was very confusing. Training MDP assumes that we repeat the situations, doesn't it? If the setup is one-shot (?) and is not any sequential decision making, it sounds more like bandit or just model-based optimization to me.
> >
> > If the target is really 'molecular optimization', why not start with model-based optimization like BO (or possibly meta-heuristic search such as GA)?
> >
> >
> > > Relation to offline RL:
> >
> > Very nice points that I don't have any answers for now. I'm also very curious about the relationship between what was presented in the paper and OPE/OPL. This is also why I think that the paper's content can be presented in a much broader setup by deeply rethinking the OPE/OPL discussions and advances.
> >
> > For now, as I described in the review, I observe a very strong similarity on the correspondence
> >
> > - Direct Method (DM) in [3][4] = plug-in estimator in the paper
> > - Importance Sampling (IS) or Inverse Propensity Score (IPS) in [3][4] = importance sampling estimator in the paper (Section 3.1.3)
> > - Doubly robust estimator in [3][4] = doubly robust performance estimator in the paper (Section 3.1.3).
> >
> > [3] Dudik, Langford, Li, Doubly Robust Policy Evaluation and Learning (ICML 2011) https://arxiv.org/abs/1103.4601
> > [4] Farajtabar et al, More Robust Doubly Robust Off-policy Evaluation. (ICML 2018) https://arxiv.org/abs/1802.03493
> >
> > and these cannot be ignored.
> >
> > My understanding is that the motivation of OPE or OPL comes from the counterfactual interest or causal ML. Evaluating a target policy p using a different policy q trained on logged (offline) data, its killer application would be recommendation
> >
> > RecSys2021 Tutorial: Counterfactual Learning and Evaluation for Recommender Systems
> > https://sites.google.com/cornell.edu/recsys2021tutorial?pli=1
> >
> > Indeed, molecular optimization or conditional molecular generation is somewhat like 'recommendation' because we will recommend any promising molecules to practitioners or domain researchers. But I felt like some aspects might be different from recommendation. Happy to have further research on this.
> >
> >
> > > Relation to offline RL 2:
> >
> > For the point why 'covariate shift' didn't exist, I think that OPE is counterfactual (or search) problem, and we need to consider the entire space X independent from probability distribution of X. I guess that this is also the case for molecular optimization or generation, because we cannot assume that similar molecules to any unidentified nice molecules are in the training dataset.
> >
> > Say X: molecular structures, y: target property. Considering the probability distribution of y would make sense, but considering the probability distribution of X does not for search problems or counterfactual problems because we can consider any input for X (we can test any X in other words).

---

### Official Review · Reviewer_zS58 · 2022-10-24

**Confidence:** 3
**Correctness:** 3
**Technical Novelty And Significance:** 3
**Empirical Novelty And Significance:** 2
**Recommendation:** 5

**Clarity, Quality, Novelty And Reproducibility:**

The clarity of the paper is ok: the main takeaways are stated pretty clearly, but I think this paper is too heavy on notation: there are many variables with different subscripts and superscripts, only a fraction of which are used multiple times throughout the paper. The implications of many of the theorems are not discussed clearly.

Novelty/originality: while I am not specifically aware of other works which break down the bias of optimization algorithms in this way, I have seen many works which aim to decompose the bias/variance of different estimators in an insightful way, particularly from the statistics literature. It would be nice to see this discussed more. In the context of this work, and the empirical work from Renz and Langevin (both cited in the paper), I would say the novelty is moderate.

Quality: I think the quality is ok. The decomposition of the bias (the key part of the paper) is a good contribution, but I think the solutions proposed are not as promising and don't seem to work well in practice (as shown in section 4). I think formulating it as a MDP also lowers the overall quality of the work.

Reproducibility: code was included, I think this is very reproducible.

**Strength And Weaknesses:**

I think the main strengths of this paper are the following:

- Focuses on relevant problem: evaluation of molecular optimization methods using predictive models is a challenging problem with no clear solution
- Decomposition of bias into two terms is insightful and potentially actionable, if the biases can be dealt with separately. I think the insight made by this paper is generally valuable
- Experiments section is interesting because they quantify these biases, albeit for a toy problem

However, I think the paper also has some concerning weaknesses:

- Formalizing the problem as an MDP in section 1 seems unnatural to me. While many molecular optimization methods do construct molecules sequentially (e.g. string-based RL), many others do not (e.g. genetic algorithms, deep generative models). The MDP formulation does not effectively capture these methods. At the same time, it seems that the only part of the MDP formulation which is essential to the authors' analysis is the reward for terminal states ($f^*$). I suggest that the authors change the problem formulation to simple optimization (e.g. solving $x^* = \max_x f^*(x)$). This is more general and also easier to understand. It would also allow the formalism to include SMILES-based methods, which I understand currently fall outside the formalism.
- Notation is often confusing, particularly when the same letters are used to represent different types of objects. For example, $\alpha_\pi$ maps a dataset to a policy, while $\alpha_f$ maps a dataset to a predictor. $J_{PI}$ maps policies and functions to performance, while $\tilde{J}$ maps datasets to final performances (with a hidden dependence on alpha). $\Delta_{PI}$ contains an implicit dependence on $f^*$. Overall I found this very confusing and constantly had to refer back to section 1-2 in order to understand what all the variables meant. I suggest the authors remove all implicit dependencies (e.g. things which depend on $\alpha$ should have $\alpha$ as a subscript) and use different letters for different types of objects.
- I think the mathematical formalism was a bit excessive, and also possibly not correct. For example, proposition 2 depends on two very technical assumptions from the appendix, which the authors intuitively describe as "$\alpha$ produces policies which only depend on the data distribution, and has a Taylor expansion. These assumptions are not really justified, and for the sake of clarity could probably be stated more simply. I actually think these assumptions might be inappropriate: for example, definition 11 defines the Taylor expansion as occurring between two Banach spaces (essentially a vector space with a norm). I don't think molecule space is a vector space since there is no obvious notion of what the addition or scalar multiplication operation might be. If molecule space is not a vector space then most of the analysis of the paper does not apply to the case of molecules. This is just one example. While I appreciate the authors attempt to be mathematically precise, I think the underlying point of this paper is fairly simple and it would be better if they tried to introduce only as much mathematical formalism as is necessary.
- Experiments are only done on a toy problem with a predictor oracle. Given that one of the main points of the paper is the bias that comes from using a ML predictor as an oracle function, I think it would be a better choice to use a non-ML oracle function as the ground-truth, for example the goal-directed benchmarks from GuacaMol (even though these are toy functions)
- The solutions proposed for reducing bias seem to only be applicable to a small class of molecular optimization algorithms. For example, bootstrap, train-test split, and policy constraint only seem applicable to RL methods trained with imitation learning (I could be wrong about this though). These limitations are not clearly stated in the paper.

**Summary Of The Paper:**

This paper analyses reasons for why the real-world performance of molecular optimization algorithms does not match the predicted performance during evaluation, which they call _bias_. Specifically, they consider the setup where a fixed dataset of molecules with labels is used to train a policy and a predictive model that is used to estimate the performance of that policy. They propose to decompose the bias into two terms: one resulting from a mismatch between the true data-labelling function and the predictive model (called _misspecification_), and one from the same dataset being used to train the predictive model and the policy (called _reuse_). The authors discuss some strategies to mitigate both of these biases, although they seem to find that they are either ineffective or hurt performance.

**Summary Of The Review:**

The main strength of this paper is the insightful decomposition of the bias of molecular optimization evaluation. However, I think the paper is made worse by its choice of a limiting MDP formalism, excessive math (which both lowers readability and may possibly be wrong, e.g. formalizing molecule space as a Banach space), and limited experiments. Altogether I think this makes the paper borderline, so I will tentatively recommend rejection at this time.

---

> ### Author Response · Authors · 2022-11-11
> **The analyses and methods can be extended to non-MDP settings.**
>
> We would like to thank Reviewer zS58 for reading our paper carefully and providing deliberate and helpful suggestions. Below are our responses.
>
> __Extension to non-MDP setting:__ By using the generator-based presentation as discussed in the general response above, we can incorporate most of the existing methods into our formalism. We would appreciate it if Reviewer zS58 could kindly suggest which of the current or the generator-based presentations is better for readers.
>
> __Assumptions:__ We would like to point out a misunderstanding; we do not assume that the molecular space is a Banach space, but we assume that the set of probability distributions over the molecular space is a Banach space, which is a common assumption. We admit that this misunderstanding is mostly due to our complex notation system, and we would like to explain each of the mathematical objects as well as their input-output relationships if they are functions.
>
> __Oracle functions:__ Smoothness of the oracle function can be relevant to the misspecification bias, but is irrelevant to the reusing bias. As the oracle function becomes less smooth, the misspecification bias tends to increase, which suggests that the actual misspecification bias could be no smaller than the one observed in our semi-synthetic experiment.
>
> The objectives in the GuacaMol benchmark are either similarity scores or the physicochemical properties like logP and TPSA that are in fact pretrained ML-based models, and we are not sure whether these objectives are better or not. We are still seeking for better options and are happy to hear any idea on it.
>
> __Solutions to reduce biases:__ Policy constraint itself is only applicable for RL agents, but its idea can be extended to the generator-based presentation discussed in the general responses, where instead of using a probability distribution induced from policy $\hat{\pi}$, namely $p_H^\hat{\pi}$, we use a general probability distribution over molecules, which we call a generator.
> The idea is extended to constrain the generator to stay close to the data distribution, and given this extension, the idea is applicable to any other methods. Other methods to reduce reusing biases (bootstrap and train-test split) are applicable to other methods without any modification.
>
> __Relation to bias-variance decomposition:__ We are grateful for pointing it out. The bias-variance decomposition usually defines the bias as the difference between an estimator $\hat{\mu}$ and its expected value $\mathbf{E}[\hat{\mu}]$, whereas we define the misspecification bias as the difference between the estimator $\hat{\mu}$ and the estimator in the limit of infinite data, $\mu^{(\infty)}$. These are equivalent if the estimator is unbiased, but in general, the estimators in our setting are biased, and the traditional bias-variance decomposition cannot be applied to our setting. Thus, our bias decomposition is related to the traditional one, but is considered to be a generalized one.

---

> > ### Comment · Reviewer_zS58 · 2022-11-16
> > **Response to authors: still have concerns about the problem setup**
> >
> > Thank you for your reply to my review; sorry that I was not able to reply sooner.
> >
> > In general I agree that:
> > - I misunderstood what was a Banach space
> > - policy constraint can be modified to be applicable to generators
> > - misspecification bias is not quite the same as bias-variance (although it is similar)
> >
> > My chief remaining concern is the applicability of the problem setup. The author's setup seems to be applicable only to methods which:
> >
> > 1. Are given a finite sample of training data points and NO access to the oracle function
> > 2. Uses these samples to train a MDP policy to generate models
> > 3. _Re_-uses these data points to train a surrogate oracle, used to evaluate the policy
> >
> > I think that not many papers actually use this setup. The authors do not provide evidence in the form of citations to convince me otherwise. I will give some examples of highly-cited/influential papers:
> >
> > - the GuacaMol benchmark [1] assumes access to the oracle function, and therefore does not train an estimator. The follow-up PMO benchmark [2] limits the number of calls to the oracle function but still allows it to be called
> > - VAE BO methods (e.g. [3]) also assume an oracle function. True, this oracle function is actually a computational estimate of a physical property, but they treat it like an oracle
> > - The commonly used SMILES RNNs (e.g. [4]) are not MDPs so the framework does not apply (although they _do_ fit into the extended generator framework). Even so, [4] does not use the same sample to train the generator and to evaluate it: they either train on the samples and evaluate the model by its ability to produce held-out known active molecules, or train a predictor and then treat the predictor like an oracle. [5], another highly-cited paper, appears to do the same thing, using the data only a single time to train a predictor.
> > - Several popular RL methods also assume an oracle [6,7,8]
> > - Optimization algorithms like BO/GAs do not correspond to an MDP or a generator: these are iterative algorithms which output new molecules given a list of previously-evaluated molecules; I believe they do not correspond to any sort of static probability distribution.
> >
> > Overall, I did not search exhaustively, but I did not find a single paper that fit cleanly into your problem definition. To me, the most important part of this definition is the use of the same samples to train an evaluation function and to train the policy. If this is not done then there is no "re-use" bias, leaving only the misspecification bias. I think the existence of misspecification bias is quite obvious (e.g. nobody with a serious ML background would expect a learned predictive model to have perfect generalization), so pointing out its existence is not a strong contribution in my opinion. The main contribution is proposing the reuse and misspecification biases as a pair, in my opinion.
> >
> > What should be the next steps? I think there are two cases to consider:
> > 1. If I am wrong about this problem setting being uncommon, the authors should state this, providing several citations to papers which use this evaluation scheme.
> > 2. If I am correct that this problem setting is uncommon, I think the authors would need to argue why this analysis is useful even if it only applies to an uncommon evaluation setting.
> >
> > All this is assuming that the authors change their presentation to the "generator" formulation. If they do not, then it seems that it is not applicable to any SMILES-based method, which are extremely common. The impact of this analysis would be greatly reduced if it does not apply to the most common representation of molecules.
> >
> > [1] GuacaMol: Benchmarking Models for de Novo Molecular Design
> >
> > [2] Sample efficiency matters: a benchmark for practical molecular optimization
> >
> > [3] Automatic Chemical Design Using a Data-Driven Continuous Representation of Molecules
> >
> > [4] Generating Focused Molecule Libraries for Drug Discovery with Recurrent Neural Networks
> >
> > [5] Molecular de-novo design through deep reinforcement learning
> >
> > [6] Optimization of Molecules via Deep Reinforcement Learning
> >
> > [7] Graph Convolutional Policy Network for Goal-Directed Molecular Graph Generation
> >
> > [8] Reinforcement Learning for Molecular Design Guided by Quantum Mechanics

---

> > > ### Author Response · Authors · 2022-11-17
> > > **Response to Reviewer zS58**
> > >
> > > Thank you very much for sparing your time to read my responses and initiating great discussions. We hope we could reach a consensus or at least we could clarify where we do not agree with each other.
> > >
> > > __Applicability of the problem setup:__ We understand that we have different views on it. Our view is that our framework can be applied to RL/generator formulation of molecular optimization using a predictor as the oracle, which includes [1-7]  Below, we provide several discussions to seek for a consensus between Reviewer zS58 and us.
> > >
> > > First of all, we consider that an evaluation method must be consistent to real use cases of the methods to be evaluated as much as possible, and therefore, let us discuss real use cases of molecular optimization methods.
> > >
> > > The first use case assumes that there exists an oracle that evaluates the property of our interest (e.g., wet-lab experiments or reliable computer simulations), and a molecular optimization method interacts with it to discover better molecules. We call this case an *online* case. The second use case assumes that there exists a dataset consisting of pairs of a molecule and its property of interest, but we have limited access or no access to the oracle due to its cost, which we call an *offline* case. The offline setting has not been explicitly stated in many papers, but a recent benchmark [7] considers this scenario, and Segler et al. (2018) [2] assume this setting (their method does not assume the oracle but only a dataset). In fact, not a few of the molecular optimization methods are developed to replace high-throughput screening (HTS) (e,g,, [2]), and they are inherently offline.
> > >
> > >  In addition, when we consider to introduce molecular optimization methods for a drug discovery process, the first step will be the offline scenario, where we investigate whether we can discover better molecules or not only from the data; if it is successful, then we integrate it to the drug discovery process, leading to the online case. Therefore, we consider the offline setting is practically important (for example, it can be integrated in the current drug discovery process), though it may not be explored in depth.
> > >
> > > Then, let us turn to the current evaluation protocols. We consider that the common evaluation protocol that substitutes a predictor for the oracle function is not consistent with the online or offline cases, and therefore, is problematic. If we assume the online case, the objective function based on the predictor is just a test function to evaluate the optimization capability, and we should not learn from the optimized molecules; if we were to learn from the optimized molecules, we had to confirm that the predictor and the oracle are close enough at the optimized molecules. But many papers show the optimized molecules and try to draw some insights from them without confirming the predictor approximates the oracle well. Then, what about the offline case? This is exactly what we have discussed in the paper; the current evaluation protocol is biased when we assume the offline case. Therefore, we consider that the current evaluation protocol needs to be improved, and we mainly focus on the offline case because of its practical importance.
> > >
> > > We consider that this huge discussion is very essential to motivate readers to consider the evaluation methods for the offline setting, and will include it in the paper.
> > >
> > > __Generator formulation:__ This is just a presentation issue and for example the work [2] already fits into our work. The initial state is fixed, and we consider one-step MDP, where the only one state transition is realized by sampling from the RNN; there is no action, and the next state is the molecule sampled from it. By considering a one-step MDP like this, the generator can be handled within the current MDP setting. Therefore, all of the theoretical results apply to the generator formulation.
> > >
> > > [1] GuacaMol: Benchmarking Models for de Novo Molecular Design
> > > [2] Generating Focused Molecule Libraries for Drug Discovery with Recurrent Neural Networks
> > > [3] Molecular de-novo design through deep reinforcement learning
> > > [4] Optimization of Molecules via Deep Reinforcement Learning
> > > [5] Graph Convolutional Policy Network for Goal-Directed Molecular Graph Generation
> > > [6] Gottipati et al., "Learning To Navigate The Synthetically Accessible Chemical Space Using Reinforcement Learning"
> > > [7] Trabucco et al., “Design-Bench: Benchmarks for Data-Driven Offline Model-Based Optimization“, arXiv:2202.08450

---

> > > > ### Comment · Reviewer_zS58 · 2022-11-17
> > > > **Clarification about generator formulation**
> > > >
> > > > In your response to my questions, you stated:
> > > >
> > > > > Generator formulation: This is just a presentation issue and for example the work [2] already fits into our work. The initial state is fixed, and we consider one-step MDP, where the only one state transition is realized by sampling from the RNN; there is no action, and the next state is the molecule sampled from it. By considering a one-step MDP like this, the generator can be handled within the current MDP setting. Therefore, all of the theoretical results apply to the generator formulation.
> > > >
> > > > I am confused. I see your point that the RNN could be viewed as a 1 step MDP, but in Appendix A example 6 it says that RNNs using SMILES strings cannot be handled in your framework. That's what I was basing my comments off of. Is the statement in the appendix incorrect? If so I'm happy to grant that SMILES can be handled in the framework. In any case, I think that the generator presentation would be clearer.

---

> > > > > ### Author Response · Authors · 2022-11-18
> > > > > **SMILES-RNN fits into our work**
> > > > >
> > > > > We are sorry to confuse you on this, and we are also sorry that our response was incorrect a little bit. Let us explain that __SMILES-RNN can be handled by our current MDP formulation__, which implies that Example 6 in Appendix A was wrong.
> > > > >
> > > > > SMILES-RNN is a generative model that receives the start token and generates SMILES tokens one by one until the end token is output or exceeding the maximum length. The resulting generative model is denoted by $p(\mathrm{SMILES}(m))$, where $m$ corresponds to a molecule, and $\mathrm{SMILES}$ function converts it to its SMILES representation. This means that SMILES-RNN defines a probability distribution over the SMILES string space.
> > > > >
> > > > > This model can be handled within the MDP formulation as follows. Let $s_\top$ be the starting special state, and we always start from it, i.e., $s_0=s_\top$. The action space is the output space of the SMILES RNN (=the SMILES string space), and we regard the SMILES-RNN as a policy, i.e., $\pi(a_0=\mathrm{SMILES}(m)\mid s_0)=p(\mathrm{SMILES}(m))$. The state transition is defined as $p(s_1 \mid s_0, a_0)=\delta(s_1=\mathrm{Molecule}(a_0))$, where $\mathrm{Molecule}$ function converts a SMILES string to a molecule. This means that $s_1$ is the molecule specified by action $a_0$ (=a SMILES string). Then, the MDP terminates at $s_1$ and we obtain the molecule. This shows that SMILES-RNN can be handled by our MDP framework.

---

> > > > > ### Author Response · Authors · 2022-11-18
> > > > > **Additional response to "SMILES-RNN fits into our work"**
> > > > >
> > > > > We have not discussed the problem of invalid molecules sampled from SMILES-RNN. Here is its clarification.
> > > > >
> > > > > We assume that we repeatedly sample from SMILES-RNN until the sampled SMILES string is valid. The sampling from $p(\mathrm{SMILES}(m))$ (and sampling from the policy $\pi(a \mid s)$) assumes this repeated sampling strategy.
> > > > >
> > > > > When drafting the original manuscript, we tried to model SMILES-RNN by $H$-step MDP ($H \gg 1$) where each time step corresponds to adding one SMILES token, and we have discovered that the repeated sampling strategy cannot be applied, and therefore, we have concluded that it cannot be handled within our framework. However, the $1$-step MDP formulation that we disclosed in this discussion allows us to use the repeated sampling strategy, and therefore, we can successfully handle SMILES-RNN within our formulation. We would thank Reviewer zS58 for the fruitful discussion that largely expands the applicability of our formulation.

---

> > > > > > ### Comment · Reviewer_zS58 · 2022-11-18
> > > > > > **Thank you for clarification, I agree SMILES RNN fits into the framework**
> > > > > >
> > > > > > I agree with you on this point, thank you for explaining.

---

> > > > ### Comment · Reviewer_zS58 · 2022-11-17
> > > > **Clarification of applicability of framework**
> > > >
> > > > Thanks for your prompt response. I also hope we can reach an agreement on this topic, or at least figure out where we disagree. I read your response several times but found it a bit confusing. Are you essentially saying that many people use offline optimization in practice, even if they do not clearly state it? I agree that many molecule optimization papers are ambiguous as to whether the method is online or offline...
> > > >
> > > > I think my key uncertainty is this question: **is the framework considered in your paper _only_ applicable to scenarios where _the same data_ is used to train a policy and a predictor of an oracle function?** In particular, what parts of your analysis apply to the following two problem settings:
> > > >
> > > > 1. The policy and value function are trained with _different_ data (e.g. the case in design bench from Trabucco et al). My impression was that your analysis did not apply here, since the same empirical distribution $\hat{G}$ was used in all places.
> > > > 2. the data is only used to train the predictor, and the predictor is used to train the policy (essentially treating it as an online problem). My understanding is that this falls outside your framework, because you consider $\alpha_\pi$ and $\alpha_f$ to be functions of $\hat{G}$, whereas in this case $\alpha_\pi$ would be a function of $\alpha_f$.
> > > >
> > > > This was the key reason why I am questioning the applicability of your framework: I think that these two problem settings are much more common than the one that you consider (see [1-7] above). To me it makes sense that your framework would not apply to these settings, since they are not _re-using_ data, so I don't see where a re-using bias would come from.
> > > >
> > > > Perhaps you can answer the following key questions for me?
> > > >
> > > > 1. Do you agree or disagree with my conclusion that your analysis is not applicable to scenarios 1-2 above? If you disagree, please explain how it it applicable, or what parts of it apply. If you agree (meaning the conclusions do not apply to very many papers), then can you make a case for why the results of your paper are interesting or important to the ML community?
> > > > 2. Can you give at least 2 examples of papers which precisely use the evaluation method you consider? I.e. using _the same_ data to train a policy and predictor, then reporting the average predictor performance. If you cannot give >= 2 examples, this would suggest to me that the setting is not realistic.
> > > >
> > > > Thanks in advance. I hope we have time for another round of communication, but if not then the answers to these questions will be very helpful for me to make a recommendation about this paper.

---

> > > > > ### Author Response · Authors · 2022-11-18
> > > > > **Re: Clarification of applicability (part 1/2)**
> > > > >
> > > > > > __Scenario 1:__ The policy and value function are trained with different data (e.g. the case in design bench from Trabucco et al). My impression was that your analysis did not apply here, since the same empirical distribution $\hat{G}$ was used in all places
> > > > >
> > > > > __Short answer.__ Our analysis is not targeted on this scenario.
> > > > >
> > > > > __A bit longer answer.__ Not targeted, but our analysis can be extended to scenario 1 straightforwardly, and we'll see a *finiteness bias*, which is a part of the reusing bias. See below.
> > > > >
> > > > > __Details.__ We would like to note, however, that it is straightforward to derive the statement like Theorem 1 for this scenario. Let $\\hat{G}$ be the train set and let ${\\hat{G}}^\\prime$ be the test set.
> > > > >  Then, the bias we care is decomposed as,
> > > > > $\\mathbb{E}_{\\hat{G}, \\hat{G}^\\prime}\\Delta_\\mathrm{PI}(\\hat{G}, \\hat{G}^\\prime)$
> > > > >
> > > > > $=\mathbb{E}_{\hat{G}, \hat{G}^\prime}[\tilde{J}_\mathrm{PI}(\hat{G},\hat{G}^\prime) - \tilde{J}_\mathrm{PI}(\hat{G},G)]$
> > > > >
> > > > > $+\mathbb{E}_{\hat{G}}\Delta_\mathrm{PI}(\hat{G},G)$.
> > > > >
> > > > > The second term is the same as the misspecification bias, and the first term is similar to the reusing bias; this bias is not caused by reusing the same sample for training and testing, and therefore, it should not be called a *reusing* bias, but the bias is due to the finiteness of the test sample $\hat{G}^\prime$, which has been discussed in footnote 1 in page 3. Let us call the latter bias a *finiteness bias* here. The finiteness bias equals zero if the model is unbiased, i.e., if the estimated predictor satisfies $\mathbb{E}_{\hat{G}}\hat{f}=f^\infty$, but as far as we are aware of, only linear models are confirmed to be unbiased, and the others are biased in general, and therefore, the finiteness bias exists.
> > > > >
> > > > > All of our analyses remain true in scenario 1 except for Proposition 2, which examines the asymptotic order of the bias, and Propositions 4 and 5, which examine the asymptotic order of the estimation accuracy. Since these are good-to-have results, we do not consider this is very limiting our contribution on scenario 1.
> > > > >
> > > > > We admit that the name of *reusing bias* is confusing because it explains the biases due to reusing the same sample *and* using a finite sample. We will rename it.
> > > > >
> > > > > > __Scenario 2:__ the data is only used to train the predictor, and the predictor is used to train the policy (essentially treating it as an online problem).
> > > > >
> > > > > __Answer:__ Our framework can handle this scenario as is.
> > > > >
> > > > > > My understanding is that this falls outside your framework, because you consider
> > > > > $\alpha_\pi$ and $\alpha_f$ to be functions of $\hat{G}$, whereas in this case $\alpha_\pi$ would be a function of $\alpha_f$.
> > > > >
> > > > > This is a little bit complicated and should have been supplemented. $\alpha_\pi$ is an arbitrary function that receives a dataset and outputs a policy (in Propositions 2, 4, and 5 we made several additional assumptions, but for simplicity, we leave them temporally). We do not assume any detail of the algorithm, and essentially *any* algorithm that has the above input-output relationship can be handled in our formulation.
> > > > >
> > > > > For example, let us discuss the online molecular optimization algorithm whose oracle is replaced with a predictor $\hat{f}=\alpha_f(\hat{G})$. Noticing that the online molecular optimization algorithm is a function that receives the MDP environment and outputs a policy, its input-output relationship can be denoted by $\pi=\alpha(\mathrm{MDP})$. The idea of replacing the oracle with a predictor can be translated into math as, substituting $\mathrm{MDP}^\prime(\hat{f})$ for $\mathrm{MDP}$, where $\mathrm{MDP}^\prime(\hat{f})$ represents an MDP whose oracle function is replaced with $\hat{f}$. Then, the input-output relationship reads $\pi=\alpha(\mathrm{MDP}^\prime(\hat{f})) = \alpha(\mathrm{MDP}^\prime(\alpha_f(\hat{G})))$, which has the same input-output relationship with $\alpha_\pi$, and therefore, this shows that the substitution strategy can be handled within our formulation.
> > > > >
> > > > > Note that the assumptions necessary for Propositions 2, 3, and, 5 are satisfied if both of the online molecular optimization algorithm and the learning algorithm of the predictor are entire, because of the compositional property of the entire function (Proposition 13).

---

> > > > > > ### Comment · Reviewer_zS58 · 2022-11-18
> > > > > > **Explanation make sense**
> > > > > >
> > > > > > Thank you very much for your answers to my questions. I agree that your framework can be applied to scenario 2, and has some implications to scenario 1. I agree that "finiteness bias" might be a better name than "re-using bias". I think this makes your analysis more applicable than I originally thought.
> > > > > >
> > > > > > Regarding works it applies to: I think it would be good to specifically mention works that your framework applies to, otherwise it may be difficult for readers to understand its implications.
> > > > > >
> > > > > > I will tentatively raise my score for now because of this + other clarifications that you have given me. Thanks so much for engaging with me during this review period.

---

> > > > > > > ### Author Response · Authors · 2022-11-18
> > > > > > > **Highly appreciated**
> > > > > > >
> > > > > > > Thank you very much for the detailed discussion, and we are happy to find a consensus between us. This discussion has been very fruitful to improve the quality of this paper, and we cannot thank Reviwer zS58 too much! We promise to update the manuscript accordingly.

---

> > > > > ### Author Response · Authors · 2022-11-18
> > > > > **Re: Clarification of applicability (part 2/2)**
> > > > >
> > > > > We are sorry for the long response.
> > > > >
> > > > > > __2 examples of papers which precisely use the evaluation method you consider:__
> > > > >
> > > > > We agree that many of the existing studies employ top-5 properties as evaluation metrics, which cannot be handled by our formulation, but there exist not a few studies that use the expectation-based evaluation metrics.
> > > > >
> > > > > The experiment reported in "Targeting the 5-$\mathrm{HT_{2A}}$" in Segler et al. (2018) seems to precisely follow the evaluation scheme we assume. It seems active molecules are reused to train a predictor and RNN, and it reports the ratio of actives, which can be written as an average.
> > > > >
> > > > > The experiment reported in "Generation of property value biased libraries with the RL system" in Popova et al. (2018) also follows our evaluation protocol. The predictor is trained by using the whole dataset based on 5-fold CV, and it is used to train a policy as well as to evaluate the policy.
> > > > >
> > > > > The work by Jin et al. (2020) regards a predictive model as an oracle ("the property predictor is then fixed throughout the rest of the training process.") and uses it for training and testing; some of the evaluation metrics are ratios, which can be handled within our expectation-based formulation.
> > > > >
> > > > > Gottipati et al. use scenario 2, which we argue is in our formulation, and they compare the performance of their method against random search in terms of the distribution of sampled molecules from the policy. Although this comparison is qualitative, when a natural quantitative comparison from it will be to compare the means of the distributions, which can be handled by our setting.
> > > > >
> > > > > - Segler et al., "Generating Focused Molecule Libraries for Drug Discovery with Recurrent Neural Networks", 2018.
> > > > > - Popova et al., "Deep reinforcement learning for de novo drug design", 2018.
> > > > > - Jin et al., "Multi-Objective Molecule Generation using Interpretable Substructures", 2020.
> > > > > - Gottipati et al., "Learning To Navigate The Synthetically Accessible Chemical Space Using Reinforcement Learning".
> > > > >
> > > > > PS. We are not very convinced of using top-5 properties as evaluation metrics, because the scores are not robust (i.e., they can change drastically when we use a different random seed), and we believe the evaluation metric should be some statistics of the distribution defined by the generator. This is one of the reasons why we employ this evaluation metric.

---

### Official Review · Reviewer_JgZe · 2022-10-25

**Confidence:** 4
**Correctness:** 3
**Technical Novelty And Significance:** 4
**Empirical Novelty And Significance:** 3
**Recommendation:** 6

**Clarity, Quality, Novelty And Reproducibility:**

The analysis is novel and the description is relatively clear. The formalism could perhaps be compressed to make way for a few more experiments.

**Strength And Weaknesses:**

Strengths

This is the first work - to my knowledge - to take a serious, theoretical take on a problem that plagues RL-based molecular optimization strategies and makes them little more than useless. The distribution mismatch between the predictor training data and the huge expressive power of RL optimizers always results in the overfit predictors being attacked by the RL optimizer. I find the partitioning of the biases and the proposed solutions above the average theoretical rigor of the works in the area.

The paper provides compelling arguments to support the intuition of why RL models struggle to invent realistic, useful molecules and how to mitigate it. The fact that mispecificaton bias seems to dominate, especially with larger datasets is also intuitive.

Weaknesses
I missed more numerical examples and real world evidence of the implications of this analysis. The proposed example is fine, but it is very minor. Just one RL, on one chemical space, for one task seems scarce. Specially because the oracle is itself a fitted function, which is likley oversmooth outside the training data (and likely somewhat overfit in its training data). The implications of this are not clear, whether what the authors report is a particularly good or a particularly bad case.

**Summary Of The Paper:**

The authors propose a theoretical take on the well-established intuition that RL-based molecular optimization approaches essentially "attack" locally-overfit predictor functions and result in bad, unrealistic molecular designs on account of these "biases" in the predictive model. They split this bias into a misspecification bias due to lack of overlap between the training data for the predictor and the actual distribution of (optimal) molecules generated by RL and a reusing bias due to the use of the same data for train and test the policy. Then, the propose solutions based on covariate shift, policy constrain, and doubly-robust performance estimators; and bias estimatio by train-test split and boostrapping respectively.
They provide an example of the quantification and reduction of these biases in a single problem

**Summary Of The Review:**

I find this work shining much needed light in the crux of RL-based molecular optimization and with higher-than-usual rigor, trying to classify the sources of bias that drive RL optimizer towards bad performance in molecular design, as well as proposing potential fixes and testing them. I miss a few more experiments (with non-ML but hard oracles [perhaps quantum chemistry?]) to further quantify the issue ~ their use of an NN-based surrogate oracle creates its own set of challenges

---

> ### Author Response · Authors · 2022-11-11
> **Using a fitted function as an oracle is not likely problematic.**
>
> We would like to thank Reviewer JgZe for supportive comments and insightful suggestions. Below are responses to the major concerns raised in the review. For the scale of the empirical studies, please consult to the general response to our thought on it.
>
> __Using a fitted function as an oracle:__ Smoothness of the oracle function can be relevant to the misspecification bias, but is irrelevant to the reusing bias. In general, as the oracle function becomes less smooth, the misspecification bias tends to increase, which suggests that the actual misspecification bias could be no smaller than the one observed in our semi-synthetic experiment. From this observation, we are not worried about substituting a fitted function as an oracle for empirical studies, though non-ML functions would be better if available. We would appreciate it if Reviewer JgZe could point out any realistic oracle function.

---

> > ### Comment · Reviewer_JgZe · 2022-11-19
> > **Acknowledged**
> >
> > I have read the authors' replies and in particular the thread with Reviewer zS58.
> > The confusion around fundamental questions like whether SMILES generative models are covered, what logP optimization entails, the overloaded notation and excessive theoretical arguments (supported on often weak assumptions), and the lack of thorough examples and benchmarks suggest to me that the paper needs more work and a thorough rewrite before it can be a useful springboard for the community. I am retaining my score for fairness sake, but the work is really _at_ the threshold, IMO.

---

> > > ### Author Response · Authors · 2022-11-20
> > > **Re: Reviewer JgZe**
> > >
> > > We thank the reviewer to follow the long discussion, and we respect your decision to keep the score.
> > >
> > > Please note that many of the confusions on SMILES generative models and applicability of our work are due to our unskilled responses in the discussion, and since we have reached a clear consensus thanks to Reviewer zS58, we believe we can update the paper so that its applicability becomes clearer.
> > >
> > > Also, note that we will update the presentation only, and the technical contents remain the same except for a wrong statement in Example 6, Appendix A. Therefore, we believe the revision is legal, meaning that the revision is not too significant.

---

### Official Review · Reviewer_EYRk · 2022-11-01

**Confidence:** 2
**Correctness:** 3
**Technical Novelty And Significance:** 3
**Empirical Novelty And Significance:** 3
**Recommendation:** 8

**Clarity, Quality, Novelty And Reproducibility:**

Novelty

The main novelty of the approach lies in the explicit decomposition of the bias of the generator into the two terms. This then allows the authors to investigate each term independently, in contrast to prior similar approaches in which the biases were intertwined.

Quality

I did not verify the derivations in detail, but they mostly follow from known results, so they seemed correct. The proposed empirical evaluation framework was technically reasonable for the setting considered in this work.

The proposed approaches to address the biases seemed reasonable.

One general limitation of the current analysis is that the “reuse” bias only arises in a limited manner. In particular, in the empirical analysis, the generator did not exhibit any bias due to sample reuse, but only due to limited training data. Thus, even in experiments designed by the authors, we are unable to evaluate whether sample reuse leads to empirical problems.

Another limitation of the current analysis is that it is not clear how much the specific observations about molecular generation generalize. While the implemented set of experiments show that the biases exist with these exact choices of distributions and models (though the “reuse” bias only shows up in a limited form as mentioned above), it is not clear whether these findings generalize. Concretely, as highlighted within the experimental design description, even within molecular generation, the hyperparameter space of predictor model class, generator model class, surrogate model approximations, etc., is huge. Thus, evaluating other molecular generation techniques within this framework would improve confidence in the generalizability of both the concepts and the specific conclusions for molecular generation.

The generalizability of the work could also be improved by discussing whether similar analyses would hold for non-MDP approaches, such as variational autoencoders. (Analysis of VAEs is hinted at in the conclusions of the paper, but it is not clear why the proposed approach would depend on a sequential generative process.)

Clarity

At some points, the notation becomes rather dense and difficult to follow. The combined performance function in Section 3.1.4 particularly jumps out. The clarity of the work could be improved by possibly condensing some of the notation, or at least adding a symbols table. Still, I appreciate that the theoretical analysis necessitates a fair amount of notation.

Otherwise, the paper is generally well structured and easy enough to follow.

The authors also provide good references to put the work in context, including connections to other related research communities like model selection and the various information criteria.

The clarity and impact of the work could be improved by highlighting which distributions (or surrogate/oracle models are needed to evaluate each type of bias. The paper could also be improved by clarifying whether the types of bias could be investigated independently when the necessary distributions/oracle models are not available for one of the biases.

Reproducibility

The work seems generally reproducible. While I did not run it, the supplementary material includes code with limited documentation. It also follows standard best practices like distributing python code as installable packages.

**Strength And Weaknesses:**

The main strength of this work lies in the explicit decomposition of the biases to allow further analysis. The empirical evaluation effectively highlights the effect of the biases and some correction methods.

Two weaknesses of the current work concern its generalizability. As discussed in more detail below, it is not clear how often these biases may empirically come up. For example, even in experiments presumably designed to demonstrate this behavior, the experiments did not show that reusing samples in the plug-in estimator and generator led to problems. Second, due to the numerous modeling choices, it is not clear to what extent the specific observations relating to molecular generation are relevant for similar settings. At the least, showing that the observations hold using multiple settings within molecular generation would improve confidence in the approach and conclusions in the paper.

**Summary Of The Paper:**

In this work, the authors observe that “plug-in performance estimators”, such as those commonly used in molecular generation and other reinforcement learning-esque settings, suffer from two kinds of bias. Misspecification bias results from policies which generate molecules far from those in the training set, while reuse bias can result from using the same data in training the plug-in estimator as well as the generator. The authors propose several approaches to correct for both types of bias. A set of empirical results demonstrate that both types of bias can arise, at least in a limited form, and that some of the proposed correction approaches can reduce the bias, though typically with a tradeoff in performance or computational cost.

**Summary Of The Review:**

Overall, such generative, RL-esque approaches have received much attention recently. This work provides a reasonable framework for analytically decoupling and evaluating two types of bias in such approaches. Thus, I believe it would be of interest to many folks at ICLR.

Still, as described above, the evaluation framework relies on a large number of choices. Thus, it is unclear if the specific conclusions from this analysis would hold in other molecular generation settings. Further, it is not clear if such distributions/oracle models would be available for other domains.

---

> ### Author Response · Authors · 2022-11-11
> **Reusing bias is not negligible in some cases, and is worth discussing.**
>
> We would like to appreciate Reviewer EYRk for valuable suggestions. See the general response above for our thought on whether our empirical studies are sufficient for our purpose or not. Below are other responses.
>
> __Reusing bias:__ We consider that the reusing bias is not negligible for some applications where the sample size is around $10^2$, though may be negligible if the sample size is more than $10^3$. Below are the reasons why we consider so.
>
> Our theoretical results suggest that the misspecification bias, $O(1)$, is often more significant than the reusing bias, $O(1/N)$, which means that the reusing bias vanishes as the sample size increases whereas the misspecification bias never vanishes. So, which bias is dominant depends on the sample size.
>
> This can be observed in Figure 1 (left). When $N=256$, the reusing bias is about a half of the misspecification bias, while when $N\geq 1024$, the reusing bias is almost negligible. In real applications, the sample size could be around $10^2$ to $10^3$ and therefore, we believe the reusing bias is not negligible in some real examples, while negligible in others, and is worth discussing in the paper.
>
> __Model choice in the experiment:__ As Reviewer EYRk suggested, experiments using a variety of models would improve the confidence of the proposed framework, still we would like to ask Reviewer EYRk to judge whether the three evidences (theoretical results, empirical result, and a connection to the penalized logP issue) are enough to support our claim that the common evaluation scheme is likely to be biased and we have to pay attention to reduce the biases. Also, for the purpose of warning the existence of biases, showing one example could be enough, because it shows that in the worst case the biases are non-negligible. See the general response above for more discussion.
>
> __Quantities necessary to estimate the biases:__ The reusing bias can be estimated without oracle models by bootstrap. The misspecification bias can NOT be estimated without oracle models, because it requires us to evaluate the oracle model $f^\star$ on the molecules discovered by the current policy, which are not in the dataset. Since the rhs of Eq. 3 is further upperbounded by $\chi^2$ divergence between $p_H^{\pi}$ and $G$, times the mean squared error of $f^\infty$, the upperbound of the misspecification bias may be estimated by it, but we have not tried it yet and left it as future work.
>
> __Extension to VAEs:__ Molecular optimization methods using VAE and Bayesian optimization can be handled by using the generator-based presentation as discussed in the general responses.

---

### Author Response · Authors · 2022-11-11
**General responses**

We would like to appreciate all of the reviewers to spend your time to review our paper and provide supportive comments and insightful suggestions. We are also very grateful if the reviewers could commit some more time to the discussion so that the present research could be better improved.

In this thread, we provide responses to the concerns raised by multiple reviewers as well as why we wish to publish this paper. We wish the reviewers and senior PCs/APCs take the followings into account when discussing the value of this paper and making a final decision.

__Value of our paper:__
Our main purpose of writing this paper is to attract more attention to the reliability issue of the performance evaluation scheme for molecular optimization, which has been mostly overlooked in the literature. As acknowledged by the reviewers, this issue is worth discussing in the community. This paper i) has theoretically explained the mechanism of the bias by decomposing it into two and has evaluated their magnitudes that the misspecification bias is $O(1)$ and the reusing bias is $O(1/N)$, ii) has demonstrated that they are not negligible at least in one example (which suggests that we always have to care about this pitfall), and iii) has related the famous issue in the penalized log P optimization to the misspecification bias. We believe these three evidences are enough (though not perfect) to support that the reliability issue is true and we have to be careful about it. Also, since this reliability issue is relevant to all of the molecular optimization studies, we believe that it will be of great importance to publicize this common pitfall to the community as soon as possible; otherwise, more and more molecular optimization methods may hack the biased evaluation scheme, leading the community to wrong directions. We would appreciate it if the reviewers could take these into account for their final decisions.

__Extension to non-RL settings:__ It is possible to extend our framework to non-RL settings by replacing the policy-induced probability distribution to a general probability distribution, which we call a *generator*. Most of the molecular optimizers in the offline setting (i.e., no access to the oracle function) can be considered as a data-dependent generator over the molecular space, $\hat{p}\coloneqq p(m; \hat{G}) \in\mathcal{P}(\mathcal{S}^\star)$.

For the RL optimizer, $\hat{p} = p_H^\hat{\pi}$, and for the VAE+BO-based optimizer, the BO module can generate optimized molecules (given a fixed dataset) by Thompson sampling.

We can evaluate its performance by $J(\hat{p}, \hat{f})\coloneqq \mathbb{E}_{M\sim \hat{p}}[\hat{f}(M)]$. This plug-in performance estimator is biased in the same way as that in the original manuscript. We haven’t employed this presentation style for clarity, but if reviewers find that this “generator-based“ presentation is clearer, we would like to revise the manuscript as such. We would appreciate it if the reviewers could suggest which is better.

__Notation issues:__ We think seriously about the notation issues and wish to improve it as much as possible. As a first step, we will add a symbol table, explaining the meaning of mathematical objects as well as their input-output relationships if available.

---

> ### Comment · Reviewer_zS58 · 2022-11-16
> **General response: value of paper**
>
> Thank you for providing an informative general response. I agree that the evaluation of molecular optimization algorithms is often problematic and deserves more attention. However, I think many papers have already pointed out that existing evaluation procedures are flawed, for example [1]. It would be good for the authors to explicitly comment on what this work adds _on top of_ existing work. Regarding the points raised by the authors, my impression is that:
>
> i) decomposing the bias is only applicable when the samples are re-used. It is unclear how common this is; when I thought about it more my concern is that this is not very common. See [my other comment](https://openreview.net/forum?id=Sh97TNO5YY_&noteId=noSYa0UgkYz) for more details.
>
> ii) it is clear that the overall bias is not negligible; this contribution seems valuable insofar as it shows that _both terms_ of the bias not not negligible. I think the authors do show this, but again it only applies in their particular evaluation setup.
>
> iii) I disagree with the author's assertion that the issues with logP are mainly "misspecification bias". While logP is indeed a computational model of actual logP which could be considered misspecified, I think most papers treat it as an oracle black-box function to optimize. That is, they imagine a setting where the _computational_ logP is the ground-truth, and _not_ an approximation of a real-world property. I think the main issue with the logP benchmark is _not_ that it doesn't exactly match real-world logP, but instead that it does not match real black-box functions which people wish to optimize in drug discovery (e.g. most large molecules will have a good logP score). Therefore I do not view this as a significant contribution to the paper.
>
> I would be curious to hear what the authors and other reviewers think about this. I think that assessing the value of this paper to the field will be an important factor for making an eventual decision to accept or reject the paper.
>
> [1] Autonomous Discovery in the Chemical Sciences Part II: Outlook

---

> > ### Author Response · Authors · 2022-11-17
> > **Re: Reviewer zS58**
> >
> > We would like to thank Reviewer zS58 for sparing your time to contribute to the review system.
> >
> > There have been a number of studies that state the penalized logP benchmark is flawed, but we consider there are two views on it. One is that optimizing the penalized logP is too easy because a longer carbon chain would increase it. The other is that optimizing the penalized log P is not what we wish to optimize in drug discovery as Reviewer zS58 suggested. We consider that both of them are the main reasons why the penalized log P benchmark should be avoided.
> >
> > Then, what about replacing the penalized log P with more realistic property of interest? If such a score can be computed by reliable simulation, then it will be a good benchmark. Otherwise, if we have to use a predictor as a proxy for it, then the molecules optimizing it suffer from the biases that we have discussed, and it is likely that the score is hacked; those optimizing the score do not necessarily optimize the property of our interest. Then, this approach (replacing the penalized log P with realistic properties) does not lead to reliable benchmarks.
> >
> > Our contribution is to theoretically explain the mechanism behind the first kind of flaw, which has not been done by prior arts [Renz et al., 2019] [Langevin et al., 2022]. This is very essential to develop a less-flawed benchmark for molecular optimization methods.
> >
> > If Reviewer zS58 and other reviewers have any other concerns, we would appreciate it if they could post them so that we can reach a consensus during the discussion phase.

---

### Author Response · Authors · 2022-11-29
**Remaining concerns**

We would like to thank all of the reviewers to commit themselves to the discussion. Thanks to the reviewers, we have found several interesting relationships to other domains, and we would like to include them in the revised paper. Since there are a number of posts in this discussion, let us summarize remaining concerns as of Nov 29, and state our responses to them. If there is anything missing, please post it in this tree. Note that we only summarize the remaining concerns; for the strengths of this paper, please refer to the original reviews, and please take both the pros and cons into consideration for the final decision.

__Relationships to OPE:__ As elaborated in [this post](https://openreview.net/forum?id=Sh97TNO5YY_&noteId=xIgp5pIeF2), our results are not straightforwardly extended to a general OPE setting, because we assume that most parts of the environment are known, which is not realistic in a general OPE setting. However, the in-depth discussion with the reviewers reveals several relationships to the OPE literature, and we would like to discuss them in the revised paper.

__Expected cumulative reward:__ As pointed out by the reviewers, the vanilla expected cumulative reward $J^\\star(\\pi)$ may not be an appropriate metric to evaluate the performance (which we have not been fully convinced of yet, and we would appreciate it if the reviewers could provide references). We agree that it is better to handle not only the expected value but also a tail statistics of the distribution as a performance metric, and we argue that our expectation-based formulation can handle tail statistics by transforming a reward $r(s,a)$ into $u(r(s,a))$, where $u\\colon\\mathbb{R}\\to\\mathbb{R}$ is a utility function as elaborated in [this post](https://openreview.net/forum?id=Sh97TNO5YY_&noteId=rtCTryUNh6s). For example, if we use $u(x)=\\exp(Cx)$ for $C>0$, the resulting expected cumulative reward prefers a larger reward than a smaller one, and it focuses on the upper-tail of the distribution. Since this transformation is straightforwardly incorporated into our formulation, we argue that our expectation-based formulation can handle the tail-statistics-based metric just by a simple reward transformation.

Interestingly, as elaborated in [this post](https://openreview.net/forum?id=Sh97TNO5YY_&noteId=Wv0xJTvbtVi), the above transformation is closely related to the entropic risk measure, and the percentile-based is closely related to the value-at-risk metric, both of which have been used to measure a tail risk of financial assets. Therefore, both metrics are proper tail statistics, which supports the validity of our expectation-based formulation (as well as the validity of the percentile-based formulation).

---

### Decision · Program_Chairs · 2023-01-20

**Decision:**

Reject

**Justification For Why Not Higher Score:**

The reviewers agreed on rejection.

**Justification For Why Not Lower Score:**

N/A

**Metareview: Summary, Strengths And Weaknesses:**


Summary:

This paper analyses reasons for why the real-world performance of molecular optimization algorithms does not match the predicted performance during evaluation, which they call bias. Specifically, they consider the setup where a fixed dataset of molecules with labels is used to train a policy and a predictive model that is used to estimate the performance of that policy. They propose to decompose the bias into two terms: one resulting from a mismatch between the true data-labelling function and the predictive model (called misspecification), and one from the same dataset being used to train the predictive model and the policy (called reuse). The authors discuss some strategies to mitigate both of these biases, although they seem to find that they are either ineffective or hurt performance.

Strengths:

- Focuses on relevant problem: evaluation of molecular optimization methods using predictive models is a challenging problem with no clear solution
- Decomposition of bias into two terms is insightful and potentially actionable, if the biases can be dealt with separately. I think the insight made by this paper is generally valuable
- Experiments section is interesting because they quantify these biases, albeit for a toy problem
- explicit decomposition of the biases to allow further analysis. The empirical evaluation effectively highlights the effect of the biases and some correction methods.

Weaknesses:

- Notation is often confusing
- The mathematical formalism was a bit excessive
- Experiments are only done on a toy problem
- The solutions proposed for reducing bias seem to only be applicable to a small class of molecular optimization algorithms
- it is not clear how often these biases may empirically come up.
- due to the numerous modeling choices, it is not clear to what extent the specific observations relating to molecular generation are relevant for similar settings

Recommendation:

This is a borderline paper. Two reviewers vote for rejection (one strongly and one midly). One reviewer votes strongly for acceptance and one tends slightly towards rejection. In the discussion, the most positive reviewer did not participate and the other reivewers agreed that the practical impact is really not there and that the theoretical results are not particularly compelling. Because of this, the reviwers agreed on rejection. I, therefore, decide to reject the paper and encourage the authors to improve the paper and resubmit to another conference.


**Summary Of Ac-Reviewer Meeting:**

In the discussion, the most positive reviewer did not participate and the other reivewers agreed that the practical impact is really not there and that the theoretical results are not particularly compelling. Because of this, the reviwers agreed on rejection.